

# Increased abyssal ocean density stratification across the Middle Pleistocene Transition

Nicola C. Thomas[1], Heather L. Ford[2], Mervyn Greaves[1], and David A. Hodell[1]

[1]Godwin Laboratory for Palaeoclimate Research, Department of Earth Sciences, University of Cambridge; Cambridge, CB2 3EQ, UK

[2]School of Geography, Queen Mary University of London; London, E1 4NS, UK

Correspondence: Nicola C. Thomas (nct30@cam.ac.uk ; nct3051@gmail.com)

**Abstract.** We report basinal and global compilations of deep-water temperature and $\delta^{18}O_{seawater}$ for the past 1.5 million years using tandem oxygen isotopic and Mg/Ca measurements of benthic foraminifera. Across the Middle Pleistocene Transition (MPT), interbasinal gradients suggest North Atlantic deep-water became colder and Pacific deep-water saltier during glacial periods after ~900 thousand years ago. Salinity in source areas increased in the marginal seas around Antarctica by decreased meltwater discharge from ice sheets and increased sea ice extent, which led to increased density stratification of the abyssal ocean. The deep ocean became a more effective carbon trap and lowered glacial atmospheric carbon dioxide, leading to expansion of continental ice sheets and longer glacial cycles. Results support a physical role for abyssal ocean stratification in explaining the MPT. Collectively, our deep ocean stacks lend support to hypotheses proposing that the MPT resulted from a progressive drawdown in glacial atmospheric $p\mathrm{CO_2}$, a conclusion that awaits verification from the Beyond EPICA–Oldest Ice core from Antarctica.

## 1 Introduction

Benthic oxygen isotope ($\delta^{18}O$) records are invaluable tools for identifying glacial-interglacial cycles of the Quaternary and providing information about changes to Earth's energy imbalance (Baggenstos et al., 2019; Bereiter et al., 2018; Grimmer et al., 2025; Shackleton et al., 2023). The interpretation of benthic $\delta^{18}O$ records is complicated, however, because the signal is dependent on both temperature and the oxygen isotopic composition of seawater ($\delta^{18}O_{seawater}$), which varies with global ice volume and local hydrographic effects (Waelbroeck et al., 2002). The deconvolution of the global deep-water benthic $\delta^{18}O$ record into its temperature and $\delta^{18}O_{seawater}$ components has been accomplished by tandem measurement of Mg/Ca and $\delta^{18}O$ (Elderfield et al., 2010, 2012; Ford and Raymo, 2020; Hasenfratz et al., 2017), with estimates of propagated uncertainty (Thirumalai et al., 2016). Other indirect approaches have also been used previously including: inverse models combining observed benthic $\delta^{18}O$ data with ice sheet models of varying complexity (Berends et al., 2021; Bintanja and Van De Wal, 2008); or, an iterative process-based approach assessing changes in relationships between the oxygen isotopic composition of ice ($\delta^{18}O_{ice}$), sea level, temperature, and $\delta^{18}O_{seawater}$ (Rohling et al., 2021). More recently changes in global mean sea surface temperatures ($\Delta$GMSST) along with proxy-based deep ocean temperatures have been used to infer changes in mean ocean



temperature (ΔMOT), which are then used to extract the deep ocean temperature and ice volume components from a benthic δ¹⁸O stack (Clark et al., 2025). Some of these approaches have led to vastly divergent interpretations of the temperature and ice volume history of the Quaternary (Clark et al., 2025).

Here we present a new 1.5-million-year-long record of North Atlantic deep-water temperature, reconstructed using Mg/Ca measured in calcite of infaunal benthic foraminifera (*Uvigerina peregrina* and *Globobulimina affinis*) from sediment cores recovered at International Ocean Discovery Program (IODP) Site U1385 from the Iberian Margin, spanning the Middle Pleistocene Transition (MPT) (Figs. 1, S1 and S2; Table 1 see Sect. 2 Materials and Methods). The MPT occurred between ~1.25 to 0.64 million years ago (Ma) when glacial-interglacial cycles in the Earth's climate system changed from dominant periods of 41 kyr to quasi-100 kyr without a commensurate change in orbital forcing (Clark et al., 2006; Pisias and Moore,

1981; Shackleton and Opdyke, 1976). The Site U1385 record from the Iberian Margin is important because of previous concerns raised about the reliability of the Mg/Ca record measured partially on epibenthic foraminifera (Sosdian and Rosenthal, 2009) at Site 607 from the central North Atlantic (Yu and Broecker, Wallace, 2010) (Sect. 3.1; Supplement). Subsequent Mg/Ca measurements of infaunal *Uvigerina* at Site 607 supported previous findings (Ford et al., 2016). The Mg/Ca-derived temperature is paired with benthic δ¹⁸O to isolate the δ¹⁸O$_{seawater}$ signal (Figs. 2 and 3). We combined (stacked)

the two North Atlantic records (Sites U1385 and 607 (Ford et al., 2016; Sosdian and Rosenthal, 2009)) and compared them with similar stacked records from the deep Pacific (Elderfield et al., 2012; Ford and Raymo, 2020) to reconstruct the basinal histories of temperature and salinity change (Sect. 2.5–2.7). The Atlantic and Pacific records are then combined with Site 1094 from the South Atlantic sector of the Southern Ocean (Hasenfratz et al., 2019) (Figs. 1, S1 and S2; Table 1) to produce global stacks.

Table 1: Site location details and modern-day deep-water properties.

| Site and Citation | Latitude | Longitude | Depth (m) | δ¹⁸O$_{sw}$ (‰) | Temperature (°C) | Salinity (psu) |
|---|---|---|---|---|---|---|
| **Iberian Margin:** **U1385** (This study; Birner et al., 2016) | 37°34.285' N | 10°7.562' W | 2578 | ~0.05 | ~3.3 | ~35 |
| **MD99-2334** (Skinner et al., 2003) | 37°48' N | 10°10' W | 3146 | ~0.05–0.10 | | |
| **MD01-2444** (Skinner et al., 2007) | 37°33' N | 10°08' W | 2637 | ~0.2 | | |
| **DSDP 607** (Sosdian and Rosenthal, 2009; Ford et al., 2016) | 41° N | 33° W | 3427 | ~0.3 | ~2.6 | ~35 |
| **Piston Core Chain 82-24-23PC** (Sosdian and Rosenthal, 2009) | 43° N | 31° W | 3406 | | | |
| **ODP 1123** (Elderfield et al., 2012) | 41°47.15'S | 171°29.94' W | 3290 | -0.05 | ~1.3 | 34.73 |
| **ODP 1208** (Ford and Raymo, 2020) | 36.1° N | 158.2° E | 3346 | ~ -0.23 | ~1.5 | 34.67 |
| **ODP 1094** (Hasenfratz et al., 2019) | 53.18° S | 05.13° E | 2807 | ~-0.05 | ~0.5 | 34.47 |





**Figure 1: Meridional profiles of the hydrographic isosurface at 2578 metres below sea level (mbsl) for oceanic deep-water $\delta^{18}O_{seawater}$ relative to site locations. (a) $\delta^{18}O_{seawater}$ isosurface at the Northeast Atlantic depth (2578 mbsl) of IODP Site U1385 (this study; yellow diamond) along with the geographic location of other sites referenced in the text: North Atlantic DSDP Site 607 (Ford et al., 2016; Sosdian and Rosenthal, 2009) (pink triangle) Southwestern Pacific ODP Site 1123 (Elderfield et al., 2012) (purple square); North Pacific ODP Site 1208 (Ford and Raymo, 2020) (lilac circle) and the Southern Ocean ODP Site 1094 (Hasenfratz et al., 2019) (red inverted triangle) located in the Atlantic sector of the Antarctic Zone. (b and c) Atlantic and Pacific meridional profiles of deep-water $\delta^{18}O_{seawater}$ (‰, SMOW) showing site locations and depths. Maps were created using GLODAPv2.2021 and GEOSECS in Ocean Data View (Schlitzer, 2019; http://odv.awi.de).**



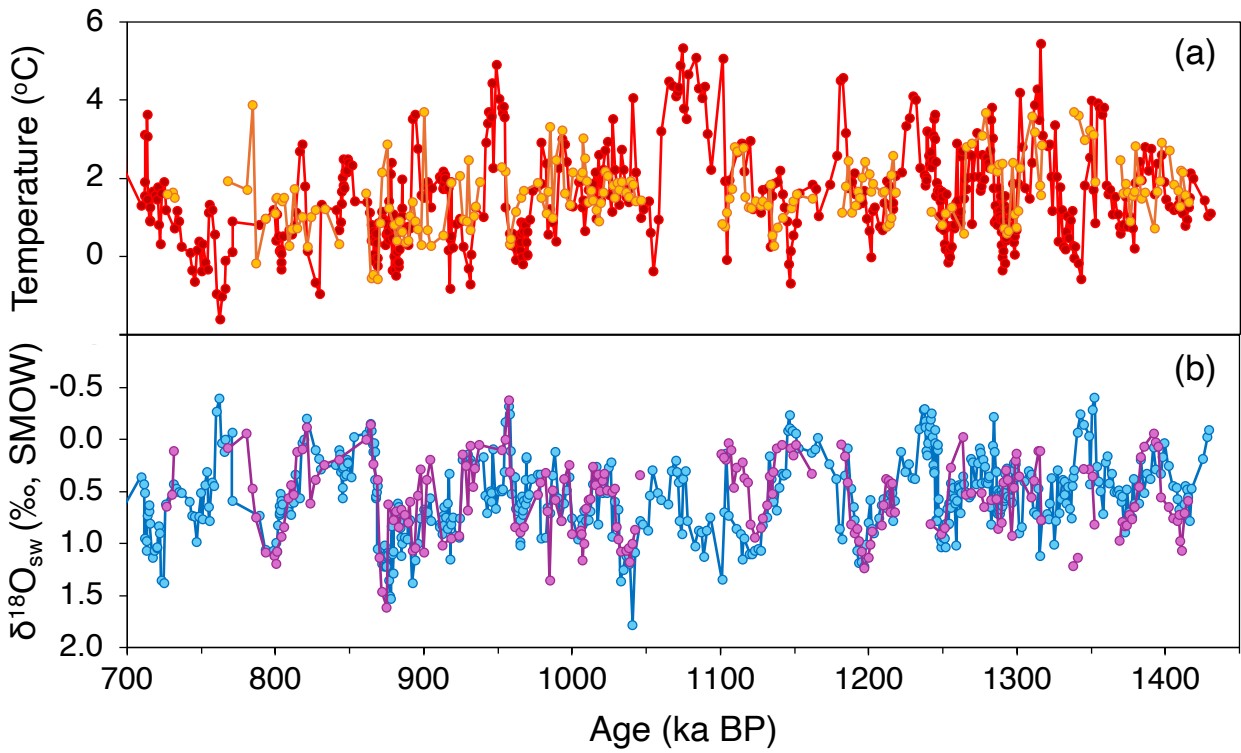

**Figure 2: Comparison of deep-water temperature and δ¹⁸O$_{seawater}$ estimates using measurements of δ¹⁸O and Mg/Ca of *Uvigerina peregrina* and *Globobulimina affinis* at Site U1385. (a) Mg/Ca-derived deep-water temperature for *Uvigerina peregrina* (red) and *Globobulimina affinis* (orange). (b) The corresponding estimated changes in δ¹⁸O$_{seawater}$ based on *Uvigerina peregrina* (blue) and *Globobulimina affinis* (purple).**

The rationale for stacking records of temperature and δ¹⁸O$_{seawater}$ is the same as for benthic δ¹⁸O (Imbrie et al., 1984; Lisiecki and Raymo, 2005); that is, to improve the signal/noise ratio and average out local signals to provide more representative regional and/or global mean signals of temperature and ice volume. Although the number of deconvolved benthic Mg/Ca–δ¹⁸O records available for stacking is few over the past 1.5 million years (Myr), some distinct differences emerge in the patterns of deep-ocean temperature and salinity change between the North Atlantic and Pacific basins.



**Figure 3: Iberian Margin deep-water Mg/Ca-derived temperature and δ¹⁸O$_{seawater}$ estimated using PSU Solver. (a) Mixed benthic δ¹⁸O, mainly *Uvigerina peregrina* otherwise *Cibicidoides wuellerstorfi* and *Globobulimina affinis* offset to *Uvigerina* by +0.64 ‰ and −0.30 ‰ respectively (black line); Numbers denote Marine isotope stage (MIS) interglacials above (odd) and glacials below (even). (b) Mg/Ca-derived deep-water temperature (light green line and markers) with 1 and 2σ error bars (dark and light grey respectively). (c) Deconvolved δ¹⁸O$_{seawater}$ (dark green line and markers; note the inverted axis) error bars as in (b). All figures show combined records from Site U1385 (this study and (Birner et al., 2016), MD99-2334 (Skinner et al., 2003) and MD01-2444 (Skinner and Elderfield, 2007).**

All study sites are deeper than 2500 m and assumed to be representative of the abyssal ocean (Figs. 1, S1 and S2; Table 1). Today, Sites U1385 (2578 metres below sea level (mbsl)) and 607 (3427 mbsl; (Ford et al., 2016; Sosdian and Rosenthal, 2009)) are bathed by North Atlantic Deep Water (NADW) and have similar seawater physical properties. Pacific Sites 1123 (3290 mbsl; (Elderfield et al., 2012)) and 1208 (3346 mbsl; (Ford and Raymo, 2020)) are located within Lower Circumpolar Deep Water (LCDW) and Pacific Deep Water (PDW) (Elderfield et al., 2012; Ford and Raymo, 2020; Insua et al., 2014), respectively, and share very similar physical properties originating from the Southern Ocean (Talley, 2013). The temperature and salinity of the deep Pacific sites are ~1.2 °C colder and ~0.3 practical salinity units (psu) lower than those sites in the North



Atlantic, reflecting the formation of relatively warm and salty NADW (Figs. S1 and S2). Correspondingly, the $\delta^{18}O_{seawater}$

values of the Atlantic sites are on average ~0.5 ‰ greater than those of the deep Pacific sites today (Fig. 1). The Atlantic and

Pacific records are merged with Site 1094 (2807 mbsl; (Hasenfratz et al., 2019)) from the South Atlantic sector of the Southern

Ocean, which is currently bathed by LCDW.

## 2 Materials and Methods

### 2.1 Chronology

Here we report Mg/Ca measured on 870 samples of *Uvigerina peregrina* and 202 samples of *Globobulimina affinis* picked

from deep-sea sediments at IODP Site U1385 located on the Iberian Margin (37°34.285' N, 10°7.562' W, 2578 mbsl) (Hodell,

2014; Hodell et al., 2015) (Figs 1, S1 and S2). Four holes drilled at Site U1385 were spliced to produce a continuous composite

section extending to 166.5 metre composite depth (mcd), equivalent to ~1.45 Ma, during Marine Isotope Stage (MIS) 47

(Hodell et al., 2015, 2023). The chronology of Site U1385 records was established by aligning the benthic oxygen isotope

($\delta^{18}O$) record to the LR04 reference stack (Hodell et al., 2023; Lisiecki and Raymo, 2005). Chronostratigraphic ages were

determined through linear interpolation between age-depth control points. All previously published records from the following

sites are presented on their original timescales: Deep Sea Drilling Project (DSDP) Site 607 (Ford et al., 2016; Sosdian and

Rosenthal, 2009); Ocean Drilling Program (ODP) Site 1123 (Elderfield et al., 2012); ODP Site 1208 (Ford and Raymo, 2020);

ODP Site 1094 (Hasenfratz et al., 2019); MD99-2334 (Skinner et al., 2003), and MD01-2444 (Skinner and Elderfield, 2007).

The high-resolution benthic $\delta^{18}O$ Prob-stack record displayed in Figs. 6–11 and S6–S9, represents the probabilistic stack that

integrates a globally distributed compilation of benthic $\delta^{18}O$ records (Ahn et al., 2017). The timescale used for the Prob-stack

is consistent with that of the LR04 reference  (Lisiecki and Raymo, 2005).

      Site U1385 benthic Mg/Ca, deep-water temperature and $\delta^{18}O_{seawater}$ records extend back to ~1.45 Ma, spanning MIS 1–

47. Mg/Ca measurements from *Uvigerina peregrina* and *Globobulimina affinis* were conducted at ~20 cm intervals, except

for two segments sampled at higher resolution: ~4 cm spacing between ~980–860 thousand years ago (ka) over MIS 27–21,

and ~5 cm over MIS 40–37 (~1300–1240 ka) (Birner et al., 2016). The corresponding mean sampling intervals are ~1820

years, ~360 years, and ~450 years, respectively. Three noticeable breaks (>15 kyr) occur in the Mg/Ca-derived temperature

and $\delta^{18}O_{seawater}$ records during key transitions. Two gaps, related to sampling limitations and low foraminiferal abundance,

occur between MIS 6–5 (~132–116 ka), and MIS 16–15 (~630–603 ka). The third is a near 30-kyr hiatus between MIS 12–11

(~430-400 ka) at Termination V (Hodell *et al.*, 2015, 2023). To produce a near continuous Iberian Margin record spanning the

past 1.5 Myr, including the MPT (Fig 3; Table.1), Site U1385 records were combined with Mg/Ca-derived temperature and

$\delta^{18}O_{seawater}$ data from the Last Glacial Maximum (LGM), measured on *Globobulimina affinis* at nearby Sites MD01-2444

(Skinner and Elderfield, 2007) and MD99-2334 (Skinner et al., 2003). Values from MD99-2334 were adjusted by +0.6 °C for

temperature and +0.15 ‰ for $\delta^{18}O_{seawater}$ to account for its deeper water setting.





## 2.2 Stable isotopes

Measurements of benthic foraminiferal $\delta^{18}O$ at Site U1385, paired (same sample) with *Uvigerina peregrina* Mg/Ca analyses to reconstruct $\delta^{18}O_{seawater}$, were predominantly obtained from the same *Uvigerina* where sufficient calcite was available prior to cleaning for Mg/Ca analysis (Barker et al., 2003) (n = 560 representing 64 % of Mg/Ca samples). When *Uvigerina* calcite was unavailable, $\delta^{18}O$ values were substituted with previously measured *Cibicidoides wuellerstorfi* (n = 309) (Hodell et al.,

2015, 2023) corrected to *Uvigerina* spp. by an offset of +0.64 ‰ (Hodell et al., 2023; Shackleton et al., 2000). To reconstruct $\delta^{18}O_{seawater}$ using *Globobulimina affinis* Mg/Ca analyses, paired $\delta^{18}O$ were obtained from corrected *Globobulimina affinis* (offset by -0.30 ‰; n = 86), *Cibicidoides wuellerstorfi* (n = 115) and *Uvigerina peregrina* (n = 1). Up to 25 and 53 specimens of *Uvigerina peregrina* and *Globobulimina affinis*, respectively, were selected from the 212–355 µm coarse fraction. Because shell calcite was also used for trace metal determination, tests were crushed between glass slides to facilitate cleaning. When

the shell material exceeded ~240 µg, a portion (>80 µg) was separated prior to cleaning and reserved for paired isotope analysis. *Cibicidoidies wuellerstorfi* $\delta^{18}O$ and carbon isotopes ($\delta^{13}C$) have previously been conducted at 1 to 2 cm resolution using 1 to 5 specimens picked from the >212 µm sediment coarse fraction (Hodell et al., 2015, 2023; Thomas et al., 2022). Isotopic measurements were made at the Godwin Laboratory for Palaeoclimate Research, Department of Earth Sciences, University of Cambridge following previously described methods (Hodell et al., 2015). Instrument precision was better than ±0.08 ‰ (1σ)

for $\delta^{18}O$ and ±0.06 ‰ (1σ) for $\delta^{13}C$.

## 2.3 Foraminiferal Mg/Ca analyses

We determined Mg/Ca on 911 samples of *Uvigerina peregrina* and 264 samples of *Globobulimina affinis* which were oxidatively cleaned following the procedure of Barker, Greaves and Elderfield (2003). The Mg/Ca-derived temperature records from *Uvigerina peregrina* and *Globobulimina affinis* show good agreement well within error (Fig. 2a) which supports

combining them. All Mg/Ca and other trace element ratios were measured using inductively coupled plasma-optical emission spectrophotometry (ICP-OES) at the Godwin Laboratory for Palaeoclimate Research (Cambridge, UK). The majority of analyses were performed on a Varian VISTA instrument (Birner et al., 2016) with the remaining ~180 samples measured using an Agilent 5100 ICP-OES. For both instruments, the intensity ratio calibration method of de Villiers, Greaves and Elderfield (2002) was applied. Interlaboratory comparison studies have validated the use of laboratory standards employed as reference

materials for foraminiferal Mg/Ca analyses (Greaves et al., 2008). Repeated measurements of standards showed instrument precision for Mg/Ca determinations to be better than 0.5 % relative standard deviation (r.s.d.) (Greaves, 2008; Greaves et al., 2008; de Villiers et al., 2002). Based on 47 sets of replicate Mg/Ca determinations, including 4 sets of triplicates, the pooled standard deviation was calculated at 0.092 mmol mol$^{-1}$ yielding an overall mean precision of 8.2 % r.s.d. (See Data Repository Table S4).





## 2.4 Data quality evaluation

Mg/Ca ratios derived from foraminiferal calcite tests can be used to estimate palaeo-ocean temperatures provided that stringent cleaning procedures are followed (Barker et al., 2003). To evaluate the efficiency of these cleaning procedures and to assess the effects of diagenesis, we analysed element/Ca ratios of co-measured trace elements—primarily Fe and Mn, but also Al, Ba, K, Na, Si, Ti and Zn (Barker et al., 2003; Elderfield et al., 2012; Greaves, 2008; Hasenfratz et al., 2017). Contamination by silicates and clay minerals can lead to significant uncertainty in measured Mg/Ca values. Therefore, the removal of contaminant silicates is the most critical step in the cleaning protocol, and its success can be evaluated through downcore covariation of Mg/Ca and Fe/Ca (or Al/Ca) records (Barker et al., 2003). Mg/Ca records from the two foraminiferal species were assessed separately, as *Globobulimina affinis* is known to exhibit elevated Mg/Ca ratios relative to other benthic foraminiferal species (Skinner et al., 2003; Weldeab et al., 2016). At Site U1385 we excluded five *Uvigerina peregrina* and three *Globobulimina affinis* samples from the Mg/Ca dataset following identification of anomalously high co-occurring Fe/Ca and Mg/Ca values (Barker et al., 2003) (Fig. S3). The remaining *Uvigerina peregrina* samples show Fe/Ca and Fe/Mg values up to 0.45 mmol mol$^{-1}$ and 0.33 mol mol$^{-1}$, respectively (or 0.79 mmol mol$^{-1}$ and 0.28 mol mol$^{-1}$ for *Globobulimina affinis*), which are considerably higher than the typical respective values of <0.1 mmol mol$^{-1}$ and <0.03 mol mol$^{-1}$ reported by Barker, Greaves and Elderfield (2003) and Elderfield et al. (2012).

Consistent with observations of *Uvigerina* spp. at Site 1123 (Elderfield et al., 2012) Mn/Ca values for *Uvigerina peregrina* at Site U1385 show a slight downcore increase, with lowest glacial values rising from ~0.012 at 60 ka to ~0.045 at ~1415 ka (Fig. S4). Although the maximum Mn/Ca value of 0.28 mmol mol$^{-1}$ is lower than the 0.36 mmol mol$^{-1}$ reported at Site 1123 (Elderfield et al., 2012) analyses suggest the potential presence of authigenic Mn-Fe-oxide coatings on the foraminiferal calcite. This interpretation is further supported by elevated Mn/Ca ratios measured in *Globobulimina affinis* specimens. However, the absence of any positive correlation between Mn/Ca and Mg/Ca suggests that this contamination does not significantly affect Mg/Ca variability. Rather, the elevated Mn/Ca values are more likely attributable to ferromanganese oxide overgrowths (Skinner et al., 2003).

Diagenetic Mn-Fe-oxide coatings commonly form on foraminiferal shells in deep-sea sediment cores used in palaeoceanographic reconstructions. In oxygenated porewater dissolved manganese is precipitated onto carbonate tests as Mn$^{4+}$ oxides, later becoming remobilized as Mn$^{2+}$ and forming Mn-rich oxide coatings when organic matter undergoes anaerobic decomposition deeper in the sediment (Barker et al., 2003; Boiteau et al., 2012). These Mn-rich coatings can substantially impact bulk Mg/Ca ratios and introduce bias into deep-water temperature reconstructions (Hasenfratz et al., 2017). Site specific studies reveal relatively consistent Mg/Mn ratios in foraminiferal coatings, ranging globally between 0.17–0.32 mol mol$^{-1}$ [see Hasenfratz et al. (2017) Table 1 and references therein]. Regionally the upper end of this range—between 0.26 to 0.32 mol mol$^{-1}$—is most applicable to the Atlantic Ocean, although individual sites may deviate notably from basin-wide averages (Hasenfratz et al., 2017). These Mg/Mn ratios can be used to correct Mg/Ca ratios associated with the calcite lattice using the following equation (Hasenfratz et al., 2017):



$$\text{Mg/Ca}_{\text{corrected}} \; = \; \text{Mg/Ca}_{\text{measured}} \; - \; \left(\text{Mn/Ca}_{\text{measured}} \; \times \; \text{Mg/Mn}_{\text{coating}}\right) \tag{1}$$


To maintain consistency, we applied a 0.32 mmol mol$^{-1}$ correction to all Mg/Ca data (Hasenfratz et al., 2017; de Lange et al., 1992) (Fig. S5). This value represents the maximum adjustment in the direction of lower temperatures and yields the least discrepancy between Site U1385 temperatures and those from the Pacific and South Atlantic. The correction has minimal influence on glacial Mg/Ca values.

Following quality control, we report Mg/Ca results for 870 samples of *Uvigerina peregrina* and 202 samples of *Globobulimina affinis*, after excluding a further 36 and 59 samples respectively, based on co-measured trace element ratios indicative of potential Mg/Ca contamination. The following criteria were used for exclusion:

- Clay contamination: Low [Ca] (<10 ppm) coupled with high Na/Ca (>5.5 mmol mol$^{-1}$), and K/Ca (>0.4 mmol mol$^{-1}$), almost always associated with elevated Mg/Ca.

- Additional indicators of clay influence: Al/Ca > 0.4 mmol mol$^{-1}$, Si/Ca > 0.4 mmol mol$^{-1}$, Ti/Ca > 0.005 mmol mol$^{-1}$, or very low Mn/Fe (< 0.1 mol mol$^{-1}$) due to elevated Fe/Ca.

- Analytical blank effects: Low [Ca] in combination with anomalously high Mg/Ca values, suggesting contamination during measurement.

**2.5 PSU Solver: Calibration equations for Mg/Ca-derived deep-water temperature and $\delta^{18}O_{\text{seawater}}$**

To calculate deep-water temperature and $\delta^{18}O_{\text{seawater}}$ estimates for Site U1385 over the past 1.5 Ma we used the Paleo-Seawater Uncertainty Solver (PSU Solver) MATLAB code (Thirumalai et al., 2016). This approach applies bootstrap Monte Carlo simulations (n = 1000) to propagate and constrain uncertainty across multiple proxies. Required input data include foraminiferal Mg/Ca and benthic $\delta^{18}O$ measurements with their respective 2 sigma ($\sigma$) analytical uncertainties, and species-specific Mg/Ca–temperature calibrations (Tables 2 and S1). Age model uncertainty was set at ±2 kyr, which represents both

the optimal resolution for combining the records of differing resolution and the mean temporal resolution of all five records.

For *Uvigerina peregrina,* we used the linear Mg/Ca–temperature calibration of Elderfield et al. (2012):

$$\text{Mg/Ca} = 1 \; + \; (0.1 \times \text{T} \,^{\circ}\text{C}) \tag{2}$$


For *Globobulimina affinis*, we applied the calibration of Weldeab, Arce and Kasten (2016):

$$\text{Mg/Ca} \; = \; 2.22 \; + \; (0.36 \times \text{T} \,^{\circ}\text{C}) \tag{3}$$



To reconstruct $\delta^{18}O_{seawater}$ we used the oxygen-isotope palaeothermometry equation (Elderfield et al., 2010, 2012):

$$T = 16.9 - 4.0 \times (\delta^{18}O - \delta^{18}O_{seawater} + 0.27) \tag{4}$$

Analytical uncertainty for Mg/Ca at Site U1385 was calculated at ±0.185 mmol mol⁻¹ (2σ), derived from the combined long-term standard error and pooled replicate error as reported in (Elderfield et al., 2012) and (Birner et al., 2016) (Tables 2 and S1). The analytical uncertainty for benthic $\delta^{18}O$ is ±0.16 ‰ (2σ).

**Table 2: PSU Solver input data for 2σ error in analytical uncertainty.**

| Site | Analytical Uncertainty (2σ) | | | |
|---|---|---|---|---|
| | Mg/Ca (mmol mol⁻¹) (long-term std + replicate error) | $\delta^{18}O$ (‰) | Species specific Mg/Ca to temperature PSU eq 1 | Lab; Citation |
| **IODP U1385** | 0.185 (Uvig) | 0.16 | Mg/Ca = 1 + (0.1 × T) ; Elderfield et al., 2012 | Cambridge: This study (Birner et al., 2016) |
| **IODP U1385** | 0.185 (G. aff) | 0.16 | Mg/Ca = 2.22 + (0.36 × T); Weldeab et al., 2016 | Cambridge: This study |
| **MD99-2334** | 0.14 (G. aff) | 0.16 | Mg/Ca = 2.22 + (0.36 × (T + 0.6)); Weldeab et al., 2016 (modified for water depth) | Cambridge: (Skinner et al., 2003) |
| **MD01-2444** | 0.14 (G. aff) | 0.16 | Mg/Ca = 2.22 + (0.36 × T); Weldeab et al., 2016 | Cambridge: (Skinner et al., 2007) |
| **ODP 1123** | 0.111 (Uvig) | 0.16 | Mg/Ca = 1 + (0.1 × T) ; Elderfield et al., 2012 | Cambridge: (Elderfield et al., 2012) |
| **DSDP 607 + Piston Core Chain 82-24-23PC** | 0.122 (Uvig): | 0.12 | Mg/Ca = 0.9 + (0.1 × T); Elderfield et al., 2012 (modified for reductive cleaning) | LDEO & Rutgers: (Ford et al., 2016) |
| | 0.202 (Cibs & Orid) | 0.12 | Mg/Ca = 1.16 + (0.15 × T); Sosdian and Rosenthal, 2009 | LDEO & Rutgers: (Sosdian and Rosenthal, 2009) |
| **ODP 1208** | 0.115 (Uvig) | 0.12 | Mg/Ca = 0.9 + (0.1 × T) ; Elderfield et al., 2012 (modified) | LDEO & Rutgers: (Ford and Raymo, 2020) |
| **ODP 1094** | 0.084 (Mpomp) | 0.14 | Mg/Ca = 0.742 + (0.119 × T); Hasenfratz et al., 2017 | ETHZ & Cambridge: (Hasenfratz et al., 2019) |






## 2.6 Stacking temperature and δ$^{18}$O$_{seawater}$ records

Stacked deep-water temperature and δ$^{18}$O$_{seawater}$ records were produced for the North Atlantic using PSU Solver estimates for Iberian Margin Sites U1385 (this study; Birner *et al.*, 2016), MD01-2444 (Skinner and Elderfield, 2007), and MD99-2334
(Skinner et al., 2003) along with Site 607 (Ford et al., 2016; Sosdian and Rosenthal, 2009). For the deep Pacific stack we used the PSU Solver output for Sites 1123 (Elderfield et al., 2012) and 1208 (Ford and Raymo, 2020) (Figs. S6 and S7). The global stacked records of mean deep ocean temperature (MDOT) and mean deep ocean δ$^{18}$O$_{seawater}$ (Figs. 4a and 5a respectively) include all the above along with ODP Site 1094 located in the South Atlantic section of the Southern Ocean (Hasenfratz et al., 2019) (Figs 1, S1, S2 and S8). The PSU Solver-derived temperature and δ$^{18}$O$_{seawater}$ estimates and their associated standard
deviations were interpolated at a regular 3 kyr interval using MATLAB's `interp` function. These interpolated values were then bootstrapped (n = 1000) by randomly subsampling with a normal probability distribution using the calculated standard deviations to produce error estimates (Figs. 4 and 5). Stacks were manually constructed by first identifying gaps in each site's original data. Mean values and associated uncertainties were then calculated across each aligned age interval. Where age gaps exceeded ~6 kyr the interpolated and bootstrapped values were excluded from the stack alignment process.

## 245 2.7 Weighted stacks

The weighted global benthic δ$^{18}$O, MDOT and δ$^{18}$O$_{seawater}$ stacks were iteratively constructed using ocean basin volume-weighting percentages from Lisiecki and Stern (2016) (see their Table S2). At each age point, the percentage of total ocean volume represented by available values from each basin was assessed. Initially each basin was weighted according to its proportional contribution to total global ocean volume, with the North and South Atlantic and Pacific Oceans together
comprising 43.6% (Table S2). However, because just over half of the global data points (n = 268) included contributions from all three basins, volume weighting was dynamically adjusted for the remaining data to accurately reflect the proportional volume each sample contributed to the total volume represented at that time step.

All stacks were smoothed using the LOWESS detrending/curve-fitting function in Acycle v2.7.1 (MATLAB) with a window of 16 kyr.






**Figure 4: Interpolated and bootstrapped Mg/Ca-derived temperature records for all sites. (a) Mean deep ocean temperature stack (MDOT) (this study; orange line and markers) with MIS numbers denoting glacial (below) and interglacial (above) stages. Mg/Ca-derived temperature records for: (b) IODP Site U1385 (this study and including other Iberian Margin records (Birner et al., 2016; Skinner et al., 2003; Skinner and Elderfield, 2007)) (light green); (c) DSDP Site 607 (Ford et al., 2016; Sosdian and Rosenthal, 2009) (light pink); (d) ODP Site 1123 (Elderfield et al., 2012)   (light blue); (e) ODP Site 1208 (Ford and Raymo, 2020) (dark purple); and (f) ODP Site 1094 (Hasenfratz et al., 2019) (yellow). Mg/Ca-derived temperature records were interpolated on a 3 kyr interval and bootstrapped. Shading represents 1 and 2σ error margins (dark and light grey respectively).**



**Figure 5: All sites interpolated and bootstrapped δ¹⁸Oseawater records. (a) Global mean δ¹⁸Oseawater stack (this study; blue line and markers) with MIS numbers denoting glacial stages (below) and interglacial stages (above). Interpolated and bootstrapped δ¹⁸Oseawater records for: (b) IODP Site U1385 (this study and including (Birner et al., 2016; Skinner et al., 2003; Skinner and Elderfield, 2007)) (dark green); (c) DSDP Site 607 (Ford et al., 2016; Sosdian and Rosenthal, 2009) (dark pink); (d) ODP Site 1123 (Elderfield et al., 2012) (dark blue); (e) ODP Site 1208 (Ford and Raymo, 2020) (purple); and (f) ODP Site 1094 (Hasenfratz et al., 2019) (dark red). All δ¹⁸Oseawater records were interpolated on a 3 kyr interval and bootstrapped. Shading represents 1 and 2σ error margins (dark and light grey respectively).**



## 3 Results

### 3.1 Comparison of Sites U1385 and 607 deep-water records

North Atlantic Site U1385 records confirm there are regional differences between the North Atlantic and South and North

Pacific. Previously, the different temperature and $\delta^{18}O_{seawater}$ signals at North Atlantic Site 607 and Southwest Pacific Site 1123 were attributed to the uncertainty in the reliability of the multispecies Mg/Ca-derived temperature record of Site 607 (Elderfield et al., 2012; Sosdian and Rosenthal, 2009, 2010; Yu and Broecker, Wallace, 2010); the high-resolution *Uvigerina* spp. record of Site 1123 was therefore considered more robust and a more global signal. However, the Site U1385 and North Pacific Site 1208 records (Ford and Raymo, 2020), both constructed using *Uvigerina* spp., confirm there are regional differences and no

one site is a completely global representation of temperature and $\delta^{18}O_{seawater}$ (ice volume) change over the MPT (Figs. 4, 5, S6 and S7).

North Atlantic Sites U1385 and 607 temperature and $\delta^{18}O_{seawater}$ records show strong covariations in amplitude and frequency over the past 1.5 Ma (Figs. 6b and 7b respectively). There are however a few notable differences between the North Atlantic Mg/Ca-derived temperature records. Greater cooling is expressed at Site 607 which cools by ~0.9 ℃ between 1500–

900 ka and ~2.2 ℃ over the past 1.5 Ma compared to respective cooling at Site U1385 of ~0.4 ℃ and 1.0 ℃. At times divergence between the two North Atlantic temperature records (e.g. between 1180–1150 ka and ~400–350 ka) corresponds to temperatures derived from *Cibicidoides wuellerstorfi* and *Oridorsalis umbonatus* at Site 607. However, these differences cannot be solely attributed to a carbonate ion effect on the epifaunal species, as between ~1400–1300 ka, temperature values derived from *Uvigerina* spp. at Site 607 are anomalously high compared to data from other species at both sites (Ford et al.,

2016; Sosdian and Rosenthal, 2009). In general, the close correspondence in both temperature and $\delta^{18}O_{seawater}$ data from the two sites supports their integration into a North Atlantic stack.



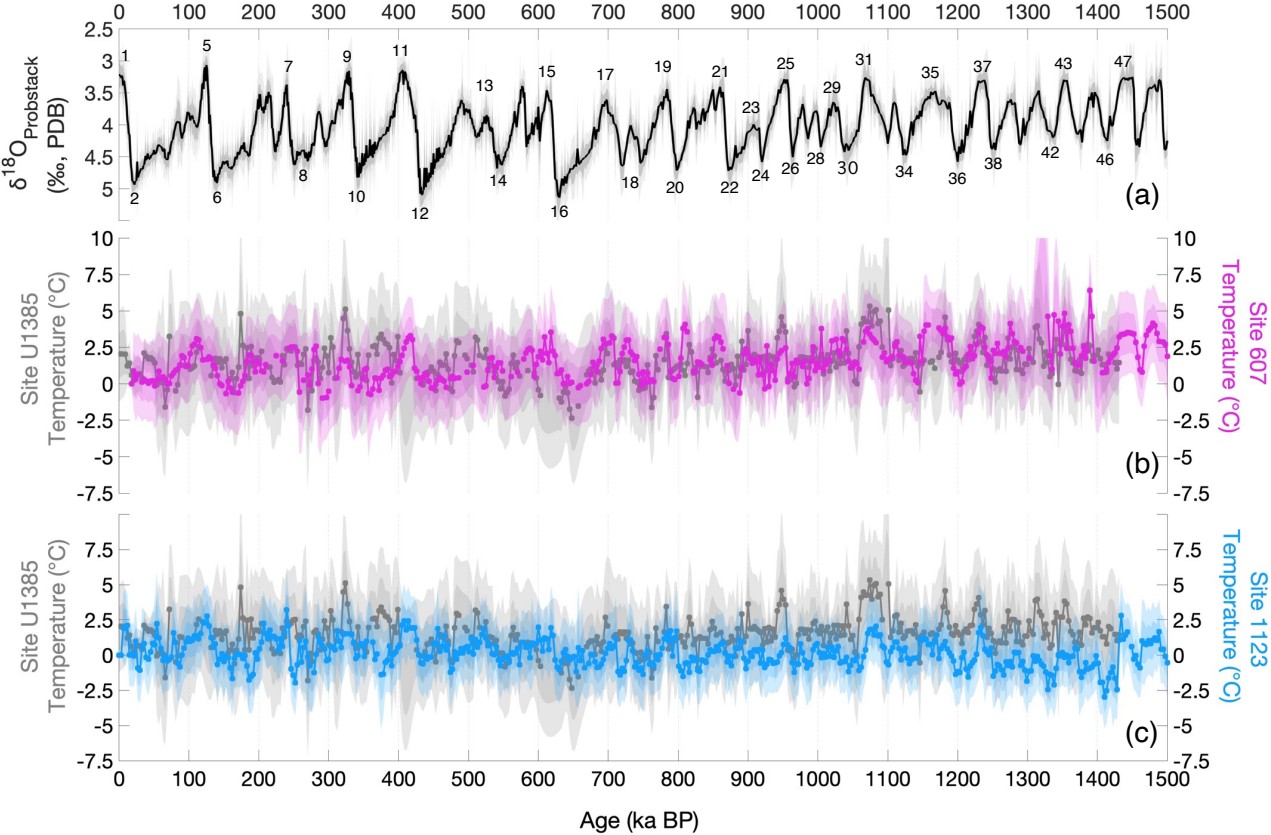

**Figure 6: Comparison of Mg/Ca-derived deep-water temperature from Site U1385 versus Sites 607 and 1123. (a) Prob-stack benthic δ¹⁸O (Ahn et al., 2017) (black line). MIS numbers denote glacial stages below and interglacial stages above. Mg/Ca-derived deep-water temperature comparisons between Site U1385 (grey) against: (b) Site 607 (Ford et al., 2016; Sosdian and Rosenthal, 2009) (pink), and (c) Site 1123 (Elderfield et al., 2012) (blue). Whereas the North Atlantic Site 607 reports a gradual decrease in deep-water temperature (of ~1–3 °C) (Ford et al., 2016; Sosdian and Rosenthal, 2009), the Southwestern Pacific Site 1123 documents relatively stable, near-freezing, glacial maxima temperatures (Elderfield et al., 2012). Same colour dark and light shading represents 1 and 2σ error margins for each parameter, respectively. Records are interpolated on a 3 kyr interval and bootstrapped.**



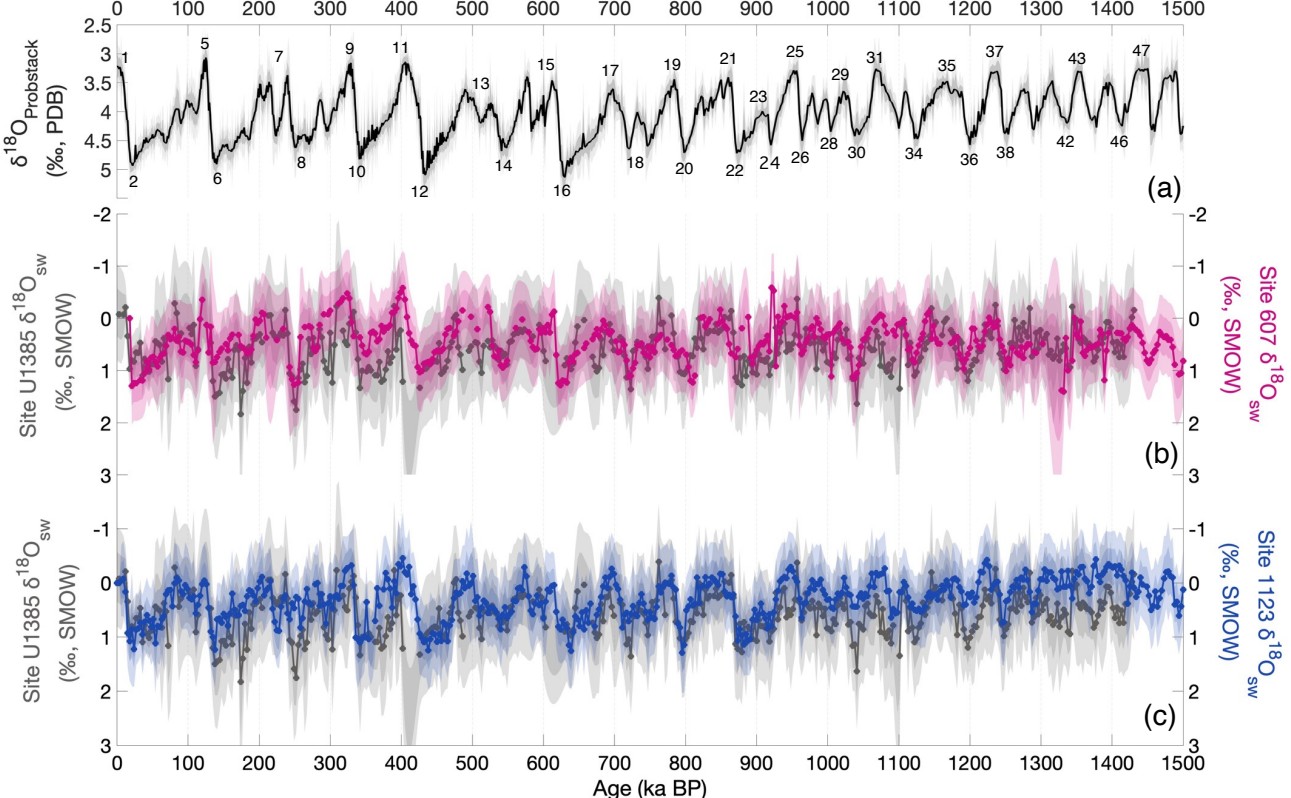

**Figure 7: Comparison of benthic $\delta^{18}O_{seawater}$ records between Site U1385 versus Sites 607 and 1123. (a) Prob-stack benthic $\delta^{18}O$ (Ahn et al., 2017) (black line). MIS numbers denote glacial (even) and interglacial (odd) stages. Deep-water $\delta^{18}O_{seawater}$ comparisons between Site U1385 (dark grey) versus: (b) Site 607 (Ford et al., 2016; Sosdian and Rosenthal, 2009) (dark pink), and (c) Site 1123 (Elderfield et al., 2012) (dark blue). Interpolated on 3 kyr intervals and bootstrapped with same colour dark and light shading representing 1 and 2σ error margins respectively for each parameter. No long-term trend in $\delta^{18}O_{seawater}$ (ice volume/salinity) is observed at Site 607 (Ford et al., 2016; Sosdian and Rosenthal, 2009) but Site 1123 reports an abrupt increase in Antarctic ice volume (salinity) ~900 ka (Elderfield et al., 2012).**

## 3.2 Stacked records of deep-water temperature and $\delta^{18}O_{seawater}$

Comparisons between Atlantic and Pacific deep-water temperature and $\delta^{18}O_{seawater}$ stacks show greater differences before than after ~900 ka. From 1500 to 900 ka (MIS 49 to 22), North Atlantic deep-water was on average ~2.5 °C warmer than the deep Pacific during both glacial and interglacial stages. Deep Pacific temperatures approach the freezing point of seawater (~-2 °C) during glacials before and after ~900 ka, whereas glacial temperatures in the deep North Atlantic cooled by more than 1 °C (Fig. 8b).

Deep ocean glacial $\delta^{18}O_{seawater}$ values increased in both oceans across the MPT reflecting increased glacial ice volume; however, the deep Pacific values increased substantially more than those of the deep Atlantic at ~900 ka (Fig. 8c). Thereafter, glacial deep-water temperature and $\delta^{18}O_{seawater}$ values from the Atlantic and Pacific were more like one another. We interpret



the greater increase in deep Pacific $\delta^{18}O_{seawater}$ as indicating an increase in salinity, in addition to the increase from expansion of continental ice sheets.

All five records were stacked to produce global signals of deep ocean temperature (>2500 m) and $\delta^{18}O_{seawater}$ change. The MDOT stack (Fig. S9c) shows a modest gradual decrease in glacial deep-water temperatures at ~1050 to 925 ka primarily due to cooling in the deep North Atlantic. In contrast, the global mean $\delta^{18}O_{seawater}$ stack (Fig. S9b) reflects an increase in continental

ice volume and salinity ~900 ka. These changes are more pronounced in ocean volume-weighted records because of a distinct warming of Pacific deep ocean temperature from MIS 24 through 22 (Figs. 9 and S10; Sect. 2.7).

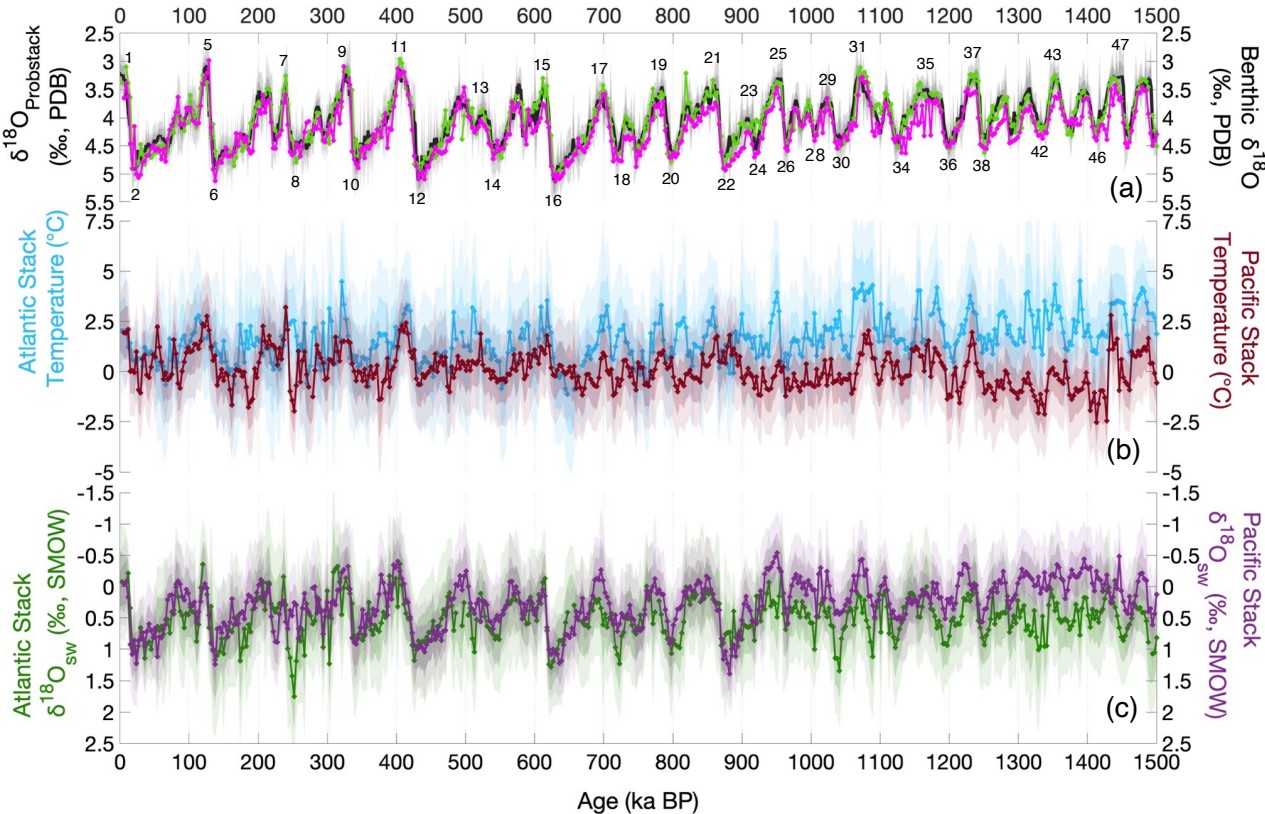

**Figure 8: Comparison of North Atlantic and Pacific Ocean deep-water temperature and benthic $\delta^{18}O_{seawater}$ stacks. (a) Prob-stack**

**benthic $\delta^{18}O$ (Ahn et al., 2017) (black line) overlain with benthic $\delta^{18}O$ records contributing to the Atlantic (this study and (Ford et al., 2016; Sosdian and Rosenthal, 2009)) (green) and Pacific (Elderfield et al., 2012; Ford and Raymo, 2020) (pink) stacks in b and c. MIS numbers denote glacial (even) and interglacial (odd) stages. (b) Deep-water temperature stacks for the Pacific (red) and North Atlantic (blue) with 1 and 2σ error envelopes shown in same colour dark and light shading respectively. (c) Deep-water $\delta^{18}O_{seawater}$ stack for the Pacific (purple) and North Atlantic (green) with error envelopes as in b. Stacks and their errors represent**

**the means of original, manually aligned, interpolated (to 3 kyr interval) and bootstrapped records (see Sect. 2 Materials and Methods).**



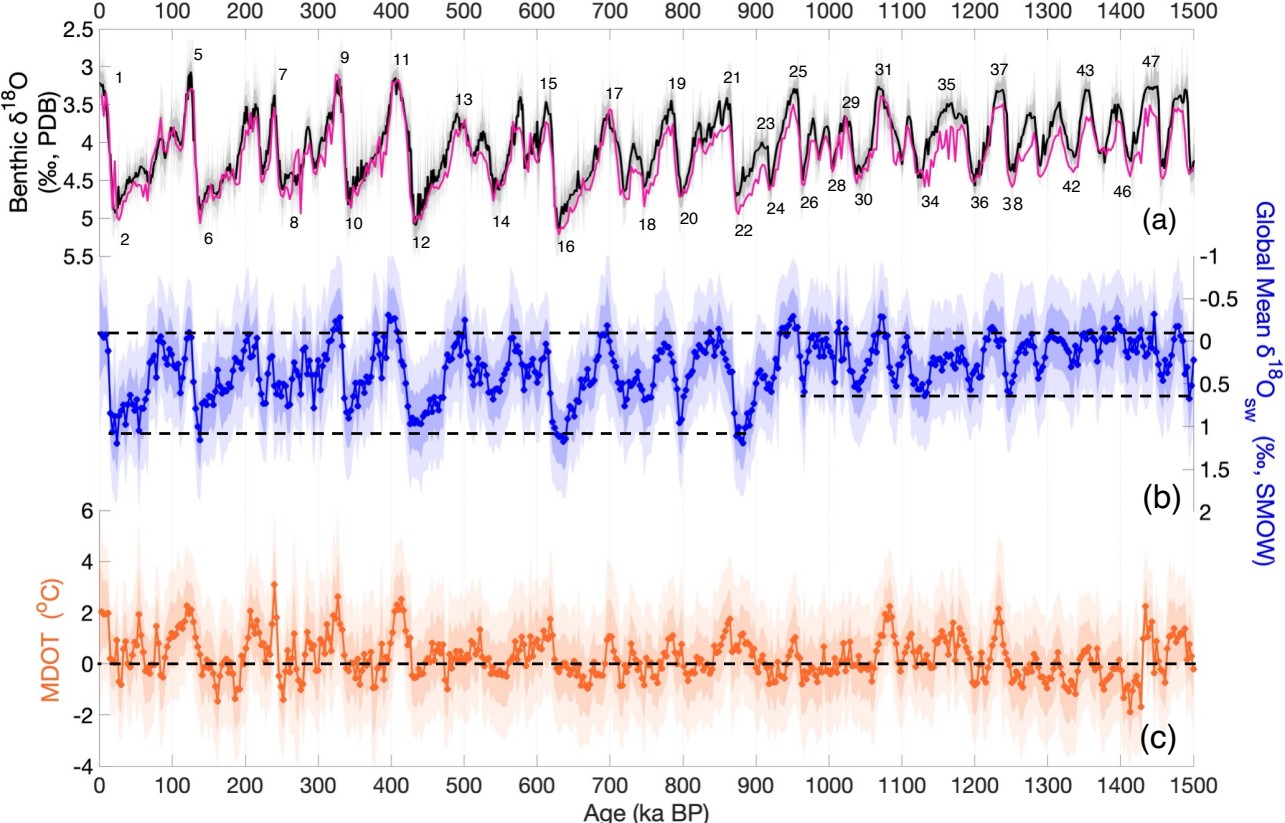

**Figure 9: Global MDOT and mean δ¹⁸O$_{seawater}$ stacks weighted by ocean basin volume. (a) Prob-stack benthic δ¹⁸O (Ahn et al., 2017)**
**(black line) overlain with the volume-weighted stack of global benthic δ¹⁸O used in this study (pink); MIS numbers denoting glacial**
**(even) and interglacial (odd) stages. (b) Global ocean volume-weighted stack of mean deep-water δ¹⁸O$_{seawater}$ (blue), showing an**
**increase in ice volume during glacial stages from ~900 ka but no significant change in interglacial values (black dashed lines). (c)**
**MDOT stack (orange) showing glacial deep-ocean temperatures have remained relatively constant (black dashed line) since their**
**lowest point during MIS 46 (~1425 ka), and with a subtle cooling through the MPT (~1050–650 ka). Compared to the non-weighted**
**records (Fig. S9), these records show a sharper increase in δ¹⁸O$_{seawater}$ (ice volume and salinity) at ~900 ka and colder deep-water**
**temperatures derived from the Pacific Ocean in the early Pleistocene. The cooling between ~1050 to 925 ka, primarily due to the**
**deep North Atlantic, is also more apparent in the weighted temperature record.**

## 4 Discussion

### 4.1 Basinal differences in abyssal temperature and salinity

Comparison of Atlantic and Pacific temperature and δ¹⁸O$_{seawater}$ records suggest an important change in the density structure

and stratification of the abyssal ocean at ~900 ka. The deep North Atlantic cooled between ~1050 to 925 ka coinciding with a

decrease in high latitude North Atlantic sea surface temperature (SST) as recorded at Site 982 on the Rockall Plateau (58° N,

16° W) (Lawrence et al., 2009) (Fig. 10b). Because deep water derived from the Southern Ocean is colder than that sourced

from the North Atlantic, the cooling may also partly result from an expansion of southern sourced water into the deep North

Atlantic. Cooling also coincides with an abrupt increase in neodymium isotopes (εNd) at Site 607 at ~900 to 870 ka (MIS 22)





previously interpreted as indicating an increased influence of Southern Component Water (SCW) in the deep Atlantic (Kim et al., 2021; Pena and Goldstein, 2014) (Fig. 10d). Several recent studies have called this interpretation into question suggesting that the deep North Atlantic was bathed in Northern Component Water (NCW) during the last glaciation (Pöppelmeier et al., 2020) and in older glacial periods of the Quaternary (Hines et al., 2024; Williams et al., 2024).

Cooling of NCW would increase its density and create a stratification inversion in the absence of a density increase of SCW (Knorr et al., 2021). The greater increase in deep Pacific $\delta^{18}O_{seawater}$ values relative to the deep Atlantic at ~900 ka suggests an increase in the salinity of the source areas for SCW formation (Figs. 8c and 10c). The $\delta^{18}O_{seawater}$ increase in the deep Pacific is attributed partly to ice volume and partly to increased salinity across the MPT. This requires a process to increase the salinity of surface waters in the marginal seas around Antarctica (e.g. Weddell and Ross Seas) where SCW forms.

Decreasing freshwater input to these source areas can be achieved by either decreasing melting of the Antarctic ice sheet (Raymo et al., 2006) and/or reduced melting of grounded ice shelves (Adkins, 2013; Miller et al., 2012). Increased sea ice formation would also raise salinity, but this process has a small and opposite effect on the $\delta^{18}O_{seawater}$ (Kim and Timmermann, 2024; Toyota et al., 2013).

The exact mechanism by which the salinity in SCW source areas was increased is not certain. Raymo et al. (2006)

suggested the East Antarctic ice sheet was terrestrial-based with melting margins before the MPT and transformed to a marine-based ice sheet afterward, which would have decreased freshwater input to the marginal seas around Antarctica. Adkins (2013) proposed that during glacial periods cooling of NADW decreased melting rates of floating Antarctic ice shelves, thereby increasing the salinity of SCW. In support of this, we observe cooling of the deep North Atlantic at the same time as inferred salinification of the deep Pacific.

Increased salinity would enhance the stratification of the deep ocean and render it a more effective carbon trap. Strengthening of SCW formation would also promote glacial deep ocean carbon storage through a standing volume effect (Skinner, 2009). Multiple proxies support decreased oxygen concentrations and increased nutrient concentrations in the deep Atlantic across the MPT (e.g. (Fig. 10e). (Farmer et al., 2019; Lear et al., 2016; Lisiecki, 2014; Thomas et al., 2022), which may have lowered the concentration of glacial atmospheric $pCO_2$ (Chalk et al., 2017; Higgins et al., 2015; Hönisch et al., 2009;

Yan et al., 2019) (Fig. 10f).




**Figure 10: Comparison of deep-water temperature, salinity, ocean circulation and atmospheric *p*CO₂ records over the last 1.5 Ma.**
**(a) Benthic δ¹⁸O Prob-stack (Ahn et al., 2017) (black line), MIS numbers denote glacial (below) and interglacial (above) stages. (b)**
**Alkenone Uᵏ'₃₇-derived, sea surface temperature (SST) record from ODP Site 982 (Lawrence et al., 2009) (dark blue line) overlaying**






## 4.2 Global versus Local Hydrographic Influences

Although only 5 long records of tandem benthic δ¹⁸O–Mg/Ca measurements exist spanning the last 1.5 Myr, we stacked the
records first with equal weighting to reconstruct global records and then weighted by the volumes of different ocean basins
(Lisiecki and Stern, 2016) (Sect. 2.7) to better estimate global averages. Whereas the unweighted glacial MDOT record shows
a slight decrease after ~900 ka, it is biased by the strong cooling of the deep Atlantic (Fig. S9c). Considering the much smaller
volume of the deep North Atlantic compared to the deep Pacific, the volume-weighted MDOT stack shows minimal change
across the MPT (Fig. 9c). Both δ¹⁸Oseawater stacks support previous findings for a step-like increase, reflecting increased
continental ice volume during most glacial periods after ~900 ka (Elderfield et al., 2012; Ford and Raymo, 2020) (Figs. 9b and
S9b).

We compare the MDOT and mean δ¹⁸Oseawater stacks with similar estimates derived by other methods. Global deep-water
stacked records agree with process-based estimates of temperature, sea level and δ¹⁸Oseawater over the past 1.5 Myr (Rohling et
al., 2021) (Figs. S11a and c), and for the last 800 ka with reconstructed Circumpolar Deep Water (CDW) (Chandler and
Langebroek, 2024) (Fig. S11b) along with estimates derived from globally compiled records (Shakun et al., 2015) (Fig. S12).
The global volume-weighted mean δ¹⁸Oseawater stack also agrees well with reconstructions of relative sea level (RSL) (Grant et
al., 2012, 2014) (Fig. 11b). Furthermore, changes in volume-weighted MDOT estimates over the last 350 ka relative to the
Holocene are supported by available mean ocean temperature (MOT) changes derived from ice-core noble gas measurements
(Grimmer et al., 2025) (Fig. 11c).



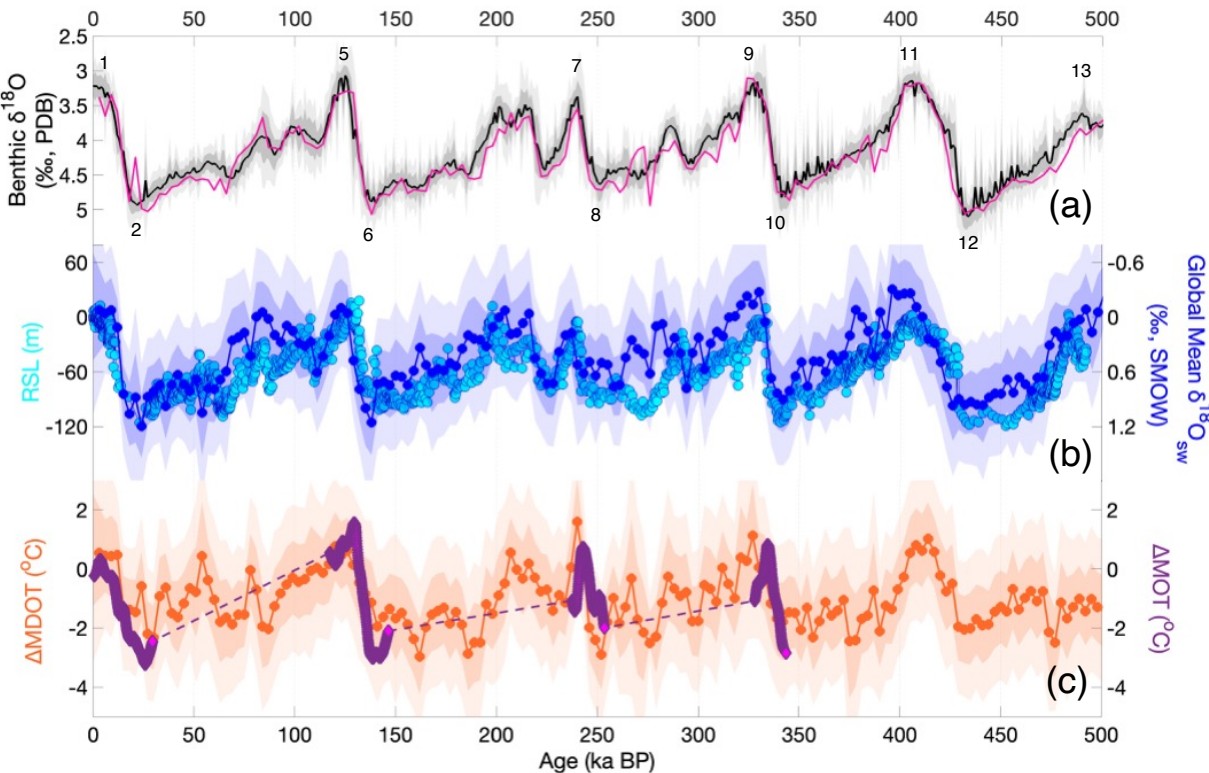

**Figure 11: Comparison of volume-weighted MDOT and global mean $\delta^{18}O_{seawater}$ stacks with ice-core noble gas measurements of MOT and Red Sea RSL. (a)** Prob-stack benthic $\delta^{18}O$ (Ahn et al., 2017) (black line) overlain with the ocean volume-weighted benthic $\delta^{18}O$ stack (this study; pink) with MIS numbers denoting glacial (below) and interglacial (above) stages. **(b)** The volume-weighted global mean $\delta^{18}O_{seawater}$ stack (this study; dark blue line with markers) versus Red Sea RSL reconstructions (Grant et al., 2012, 2014) (light blue circles) scaled so that $\delta^{18}O_{seawater}$ of 1.0 ‰ is equivalent to 10 m RSL. **(c)** The change in volume-weighted MDOT ($\Delta$MDOT) relative to the Holocene (1.5 ºC (Goudsmit-Harzevoort et al., 2023; Rohling et al., 2021, 2022)) (this study; orange line with circles) overlain by $\Delta$MOT derived from noble gas measurements in ice-cores (Grimmer et al., 2025) (purple line with pink diamonds).

Results differ substantially from estimates of changes in deep ocean temperature (>2000m) and mean $\delta^{18}O_{seawater}$ constrained using $\Delta$MOT inferred from proxy-based deep ocean temperatures and changes in global mean sea surface temperatures ($\Delta$GMSST) (Clark et al., 2024, 2025). For the period from 1.5 to ~0.9 Ma prior to the MPT, warmer deep-water temperatures and higher $\delta^{18}O_{seawater}$ (greater ice volume) are calculated compared to our estimates. A large drop in deep ocean temperature (>2000 m) and no significant increase in ice volume are inferred at ~900 ka (Fig. 12). In contrast, the global MDOT stack shows near-freezing deep water temperatures during glacials both before and after the MPT with little evidence of cooling, except in the North Atlantic. The Clark et al. compilation is strongly biased towards the deep North Atlantic (predominantly Sites 607 and U1313) with only ~12 data points from the Pacific used to constrain deep ocean temperature (>2000 m) (Clark et al., 2025; Lear et al., 2003). Site U1385 does not show strong cooling between 1.5 to ~0.9 Ma (Figs. 2A, 3B, 4, 6, and S6; Sect. 3.1) but the record from Sites 607 and U1313 is unlikely to represent global mean changes and is complicated by multi-species measurements (Ford et al., 2016). Instead, our findings support previous interpretations for near-



freezing glacial bottom water temperatures in the Pacific throughout the period from 1.5 to ~0.9 Ma with no significant change

across the MPT (Elderfield et al., 2012; Ford and Raymo, 2020).

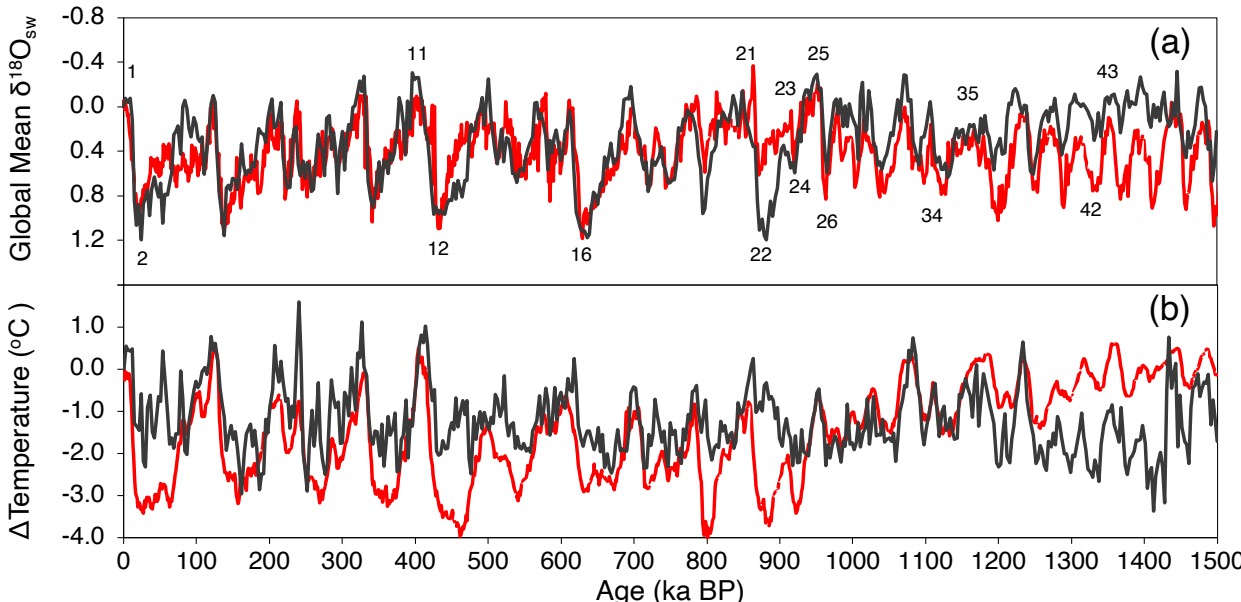

**Figure 12: Comparison of MDOT and global δ¹⁸O_seawater against similar process-based stacks. (a) Volume-weighted global δ¹⁸O_seawater stack (this study; black) against δ¹⁸O_seawater derived from the deconvolution of the benthic δ¹⁸O_ProbStack using inferred changes in**

**MOT (Clark et al., 2025) (red). Numbers denote MIS with glacial below and interglacial above. (b) Volume-weighted ΔMDOT (this study; black) presented as the change from the preindustrial value (1.5 ºC) (Goudsmit-Harzevoort et al., 2023), versus ΔMOT inferred from ΔGMSST and proxy-based deep-water temperature (Clark et al., 2025) (red).**

The interpretation of the MDOT history and global versus local/regional changes in global volume-weighted δ¹⁸O_seawater

has important implications for ice volume changes across the MPT (Fig. 12 and S13). Clark et al. (2025) infer little to no change in ice volume across the MPT which is contrary to most previous work. They discounted records of deep Pacific Ocean temperature and δ¹⁸O_seawater from Pacific Sites 1123 and 1208 (Elderfield et al., 2012; Ford and Raymo, 2020) claiming they are unrepresentative of global values and reflect regional hydrographic changes. While we agree there is a salinity component to the increase in the glacial deep Pacific δ¹⁸O_seawater signal, there is also a global increase reflecting greater ice volume in most

glacial periods after 0.9 Ma.

We acknowledge limitations to our study which underscores how comparable studies can improve upon this work in the future. The small number of tandem Mg/Ca–benthic δ¹⁸O records available to stack leads to uncertainty in how well they

represent global or basin-wide averages. The sites are all >2500 mbsl and a greater depth range would provide a better comparison with noble gas-based MOT estimates across the MPT.





## 5 Conclusions

The causes of the Middle Pleistocene Transition have been sought since the frequency and amplitude change of glacial-interglacial cycles were first recognized over 50 years ago (Pisias and Moore, 1981; Shackleton, Nicholas John and Opdyke,

1973). The MPT involved a fundamental change in the interaction of deep-ocean circulation, $p$CO$_2$, and the cryosphere that permitted expansion of continental ice sheets (Chalk et al., 2017; Elderfield et al., 2012; Ford and Raymo, 2020; Higgins et al., 2015; Hönisch et al., 2009; Pena and Goldstein, 2014; Yan et al., 2019). Here we demonstrate the MPT involved changes in the physical properties of the abyssal ocean, including a cooling of the deep North Atlantic and salinification of the deep Pacific. This resulted in increased density stratification of the deep ocean in glacial periods after ~900 ka. The abyssal ocean

became a more effective carbon trap that was fuelled partly by increased export production and surface stratification in the Southern Ocean (An et al., 2024; Billups et al., 2018; Ferrari et al., 2014; Hasenfratz et al., 2019; Jaccard et al., 2013; Martínez-Garcia et al., 2011; Qin et al., 2022; Rodríguez-Sanz et al., 2012; Starr et al., 2021; Weber et al., 2022).

Increased ocean carbon storage would have lowered glacial atmospheric $p$CO$_2$ (Chalk et al., 2017; Higgins et al., 2015; Hönisch et al., 2009; Yan et al., 2019), thereby permitting the ice sheets to grow larger and making them less susceptible to

complete deglaciation with each obliquity cycle (Huybers, 2006; Huybers and Wunsch, 2005). Increased salinity of the deep Pacific implies decreased freshwater input into the source areas of SCW formation around Antarctica and increased sea ice extent. It is unclear whether this was brought about by a transition from terrestrial melting margins to marine-based ice sheets on Antarctica (Raymo et al., 2006) and/or decreased melting of floating ice shelves by a cooling of NADW (Adkins, 2013).

The stacked mean $\delta^{18}$O$_{seawater}$ record supports previous findings of an increase in glacial ice volume at ~900 ka and an

intensification of the glacial cycles (Elderfield et al., 2012; Ford and Raymo, 2020). The anticipated recovery of a continuous Antarctic ice core across the MPT (Wolff et al., 2022) will provide critical information, especially greenhouse gases and MOT, needed to test many of the inferences made herein on the basis of the stacked MDOT and $\delta^{18}$O$_{seawater}$ records.



**Data availability.** All data, not previously reported, has been deposited at the World Data Center PANGAEA repository
(www.pangaea.de.) and will be publicly available when the paper is accepted and a DOI issued
Previously reported benthic $\delta^{18}O$ for IODP Site U1385 (Hodell et al., 2023) are archived at:
(https://doi.org/10.1594/PANGAEA.951401)

**Supplement link.**

**Author contributions.** NCT and DAH designed the project. NCT conducted all analyses in collaboration with: DAH for stable isotopic analyses, MG for trace element analyses, and HLF for PSU Solver analyses. NCT and DAH wrote the original draft, and all co-authors contributed to the final manuscript.

**Competing interests.** One of the (co-)authors is a member of the editorial board of Climate of the Past. The authors declare that they have no other competing interests.

**Disclaimer.** Publisher's note:

**Acknowledgements.** We thank all members of the Godwin laboratory for Palaeoclimate Research in the Department of Earth Science, University of Cambridge, UK for technical assistance: J. Booth and M. Mleneck-Vautravers for selection and preparation of samples for stable isotope analysis; J. Rolfe and J. Nicolson for performing stable isotope analyses; J. Nicolson and M. Mleneck-Vautravers for much of the preparation of samples for trace element spectrophotometry and conducting ICP-OES analyses. The International Ocean Discovery Program (IODP) provided samples used in this research. We are thankful to the crew of the *JOIDES Resolution* during IODP Expedition 339 including the drilling crew, ship's crew, and scientific and technical staff who made core recovery from Site U1385 possible.

**Financial support.** The Natural Environment Research Council (NERC) provided funding to DAH to collect IODP and JC089 samples and for all laboratory and analytical costs. NERC grants NE/R000204/1 and NE/K005804/1 (to DAH). NCT has been privately funded by R. M. Thomas.

**Review statement.** included by Copernicus.

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
