# Peer review of "Increased abyssal ocean density stratification across the Middle Pleistocene Transition"

_EGUsphere, 2025_

## Author Comment (AC1)

Thomas et al. present combined benthic foraminiferal Mg/Ca and oxygen isotope data from IODP Site U1385 at the Iberian Margin to reconstruct $\delta^{18}O_{sw}$ (seawater oxygen isotopes) covering the past 1.5 Myrs. The record is compiled with similar datasets from the Atlantic and Pacific sectors of the Southern Ocean as well as from the North Pacific, allowing the authors to produce a deep ocean $\delta^{18}O_{sw}$ and temperature stack that reflects global changes. The study uses a (to my knowledge) novel approach by applying a volume-weighted averaging scheme to calculate the global $\delta^{18}O_{sw}$ stack, that accounts for the relative sizes of the major ocean basins. This method is contrasted with previous approaches that did not include ocean volume-based corrections, providing new insights into the interpretation of global signals. The main findings of the study indicate increased salinity in the deep Pacific and reduced deep-water temperatures in the Atlantic during glacial periods after 900 ka, compared to the interval between 1.5 and 0.9 Ma. The authors interpret these results as evidence for enhanced glacial water mass stratification in the Pacific after 0.9 Ma, leading to a greater sequestration of atmospheric carbon dioxide in the deep ocean, as well as increased salinity in surface waters within regions of southern component water formation around the Antarctic margins.

The study draws on an extensive dataset, and the methodology appears sound and appropriate to the study's objectives. The resulting dataset has the potential to make a valuable contribution to our understanding of global ocean temperature, salinity and ice volume changes across the Mid-Pleistocene Transition (MPT). However, the strength of the methodological and analytical work is not matched by the clarity and depth of the data presentation and interpretation. In its current form, the manuscript does not fully exploit the scientific potential of the dataset, and several key aspects of the analysis and discussion require revision and elaboration. Below I outline my major and minor points of criticism in the hope that the authors find them useful in improving and revising their manuscript.

We thank the Reviewer for their detailed review of our paper and appreciate their helpful comments.

We have modified the introduction and discussion to emphasise the implications of our results to understanding changes in global ocean temperature, salinity and ice volume across the Mid-Pleistocene Transition (MPT).

In the following, I will first outline several broad, major concerns that affect the overall interpretation and structure of the manuscript. I will then provide detailed, section-by-section comments and suggestions aimed at improving the scientific robustness and overall comprehensiveness of the manuscript.

1. The manuscript warrants more transparency in the many nuances of the interpretation and should clearly outline the limitation of the data interpretation. For instance, the authors base their conclusion of an enhanced density stratification throughout the entire deep Pacific after 900 ka (compared to 1.5–0.9 Ma) primarily on data from only one Site in the South Pacific and one in the North Pacific. This interpretation is presented as a definitive conclusion, rather than as a working hypothesis supported by limited regional evidence. Potential variations in salinity throughout the entire water column—and their corresponding effects on vertical stratification—are not discussed, and possible regional hydrographic differences are insufficiently addressed in my opinion.

We acknowledge the spatial coverage in the Pacific is limited, and that our interpretation of enhanced deep Pacific density stratification after ~900 ka should be framed more cautiously. In the revised manuscript, we will explicitly rephrase this interpretation as a working hypothesis supported by available North and South Pacific sites, rather than as a definitive basin-wide conclusion.

We will also expand the discussion to acknowledge potential vertical and regional variability within the Pacific water column and clarify that our conclusions pertain to the sampled deep-water masses rather than the entire Pacific basin. Additional text will be added to discuss how salinity changes at different depths, as well as unresolved regional hydrographic differences, could influence the reconstructed stratification signal.

These revisions will improve transparency regarding the limitations of the dataset and the scope of the conclusions that can be drawn.

Elderfield et al. (2012) argued that more than half of the flux of cold bottom water entering the major ocean basins transits the Southwest Pacific as part of the Pacific Deep Western Boundary Current (DWBC) (Elderfield et al., 2012), consistent with earlier work by McCave et al. (2008) and Whitworth et al. (1999). This view is supported by evidence from the Last Glacial Maximum (Insua et al., 2014) and, more recently, across the MPT (Li et al., 2026). Although limited to two sites, it is reasonable to assume that Pacific Sites 1123 and 1208 may more broadly reflect conditions in the deep Pacific. Similar records from other sites are needed to test if this assumption is correct.

2. Throughout the manuscript, several sections lack a clear explanation, and important arguments are presented in a way that they appear somewhat ambiguous. This occurs in multiple places and affects the overall clarity of the manuscript (e.g. comments on lines 42, 127, 172-177, 191, section 4.1). I recommend using more precise wording in many sections to avoid confusion.

As suggested by the reviewer we will revise the highlighted passages (e.g. lines 42, 127, 172–177, 191, and Section 4.1) to improve clarity, reduce ambiguity, and ensure that key arguments are stated explicitly and consistently.

3. In the abstract and discussion, the authors attribute the observed increase in deep Pacific salinity after 900 ka to changes in the source regions of Southern Component Waters (SCW), i.e., Antarctic margin regions (e.g., the Weddell and Ross Seas). However, this interpretation (i.e. a key finding in the abstract) is not substantiated by any evidence presented in the study. In my view, the manuscript provides insufficient discussion of existing literature and data that would support such large-scale changes in SCW source regions during this interval. Revision is needed here. Moreover, as PDW is a mixture of several water masses, the possibility of temporal changes in PDW end-member composition—and their influence on reconstructed salinity—should be addressed, to support the suggested changes in SCW across the MPT. The authors could consider potential variations in inter-basin connectivity within the Southern Ocean, e.g. using the Antarctic Circumpolar Current strength reconstruction of Lamy et al. (2024), which could help assess whether large-scale circulation

changes—rather than solely local shifts in AABW formation—may contribute to the observed Atlantic–Pacific differences across the MPT.

We agree with the Reviewer that the attribution of increased deep Pacific salinity after ~900 ka requires more careful framing and broader contextualisation. In the revised manuscript, we present changes in Southern Component Water (SCW) source regions as a working hypothesis rather than a definitive conclusion.

We will substantially expand the discussion to integrate recent and relevant literature that bears on Southern Ocean processes across the MPT, including evidence for changes in Antarctic ice-sheet configuration, abyssal stratification, and deep-ocean circulation (McKay et al., 2012; An et al., 2024; Wirths et al., 2025; Li et al., 2026; Scherrenberg et al., 2025). In addition, we now explicitly consider the multi-sourced nature of Pacific Deep Water and discuss how temporal changes in PDW end-member composition could influence reconstructed deep Pacific salinity.

To address the possibility that large-scale circulation changes, rather than only local changes in AABW source areas, may also have contributed to the observed Atlantic–Pacific contrasts, we have incorporated discussion of inter-basin connectivity within the Southern Ocean. In particular, we draw on recent reconstructions of Antarctic Circumpolar Current (ACC) strength (Lamy et al., 2024) to assess whether changes in ACC strength may have modulated deep-ocean exchange and contributed to the large-scale patterns observed across the MPT including in the Southern Ocean (Williams et al., 2024).

4. Throughout the manuscript, statistical analyses are insufficiently applied when comparing different datasets and time intervals. Statements referring to "significant" differences are frequently made without presenting the results of any formal statistical tests. To substantiate these claims, appropriate statistical analyses (e.g., t-tests, ANOVA) should be conducted and reported. Including quantitative measures of variability and statistical significance would greatly strengthen the robustness of the interpretations and allow readers to assess whether observed differences between regions or time periods are statistically significant.

We agree with the Reviewer that statements referring to "significant" differences should be backed by formal statistical testing.

Where necessary, in the revised manuscript, we will apply appropriate statistical tests (e.g. t-tests or non-parametric equivalents, where applicable) when comparing different datasets, and we will report the results explicitly including quantitative measures of variability and uncertainty to allow readers to assess the robustness of observed differences. Where statistical significance cannot be demonstrated, we will revise the text to avoid overinterpretation.

5. In my opinion, the introduction lacks a clear structure and does not fully establish the central research question. Integrating relevant key studies (e.g. Quin et al., 2022)—many of which are discussed in the discussion (e.g. Hines et al., 2024; Pena and Goldstein 2014, Farmer et al., 2019; Lear et al., 2016) —more prominently at the start would help frame the topic, clarify existing knowledge gaps, and better motivate the study within its scientific context. Moreover, a concise overview of the main

hypotheses proposed to explain the MPT—including e.g. the regolith hypothesis (e.g. Clark and Pollard, 1998, Clark et al., 2006), the expansion of North American and/or East Antarctic ice sheets (e.g. An et al., 2024; Bintanja and van de Wal, 2008; Raymo et al., 2006), or feedback mechanisms involving changes of the marine carbon cycle (e.g. Chalk et al., 2017; Willeit et al., 2019)—would provide important context and more clearly situate the study within the broader landscape of MPT research.

In the revised manuscript, we have restructured the Introduction to provide a clearer conceptual framework and to better motivate the study.
Specifically, the Introduction now opens with a concise overview of the MPT and summarises the principal hypotheses proposed to explain it.

Key studies previously introduced later  in the Discussion (e.g. Lear et al., 2016; Farmer et al., 2019; Qin et al., 2022; Peña and Goldstein, 2014; Hines et al., 2024) are now integrated into the Introduction to clarify existing knowledge gaps and to position our study more clearly within the current literature.

We then explicitly articulate our central research question: *Was there a shift in abyssal ocean density structure across the MPT?* This leads directly into our use of benthic $\delta^{18}O$ and Mg/Ca records to disentangle the temperature and $\delta^{18}O_{seawater}$ components of density stratification and to evaluate their implications for deep-ocean circulation across the MPT.

6.  Overall, the figures would benefit from improved clarity and more explicit information to help the reader follow the comparisons and interpretations of the many timeseries presented. It would be helpful to include the location of each record directly in the figure, for instance by labelling the regions (e.g., Pacific, Southern Ocean, Atlantic…) in the figures. For the Iberian Margin record, it should be clearly indicated which data were generated in this study and which were compiled from previous publications (different symbols or colours). Finally, it would be beneficial to show key time intervals discussed in the manuscript at higher temporal resolution, rather than referring exclusively to the long composite records. This would allow for a clearer evaluation of the main features and transitions emphasised in the text. If feasible, reducing the overall number of figures may improve the readability of the manuscript, as the large number of time series and figures can be somewhat overwhelming for the reader.

    We appreciate the Reviewer's observations and will revise the figures to improve clarity and readability.

    We will clearly indicate which Iberian Margin data were generated in this study and which were compiled from previously published records, using distinct symbols and/or colours. The regional origin of each record (e.g., Atlantic, Southern Ocean, Pacific) will be explicitly labelled within the figures.

    We will also reduce the number of figures where possible by combining or eliminating redundant panels. In addition, we will include more detailed plots over key intervals discussed in the text (e.g., pre-1.2 Ma, 1.2–0.65 Ma, and 0.65–0 Ma for the ocean stacks) to better highlight the main features and transitions contained in the records.

7. Salinity on the Practical Salinity Scale (PSS-78) is a dimensionless quantity, and the use of the unit "PSU" (Practical Salinity Unit) is therefore incorrect. The authors should avoid using "PSU" throughout the manuscript. Instead, it should be stated upon first mention that all salinity values are reported on the Practical Salinity Scale (PSS-78) and are thus unitless. It would be preferable to adopt the TEOS-10 (Thermodynamic Equation of Seawater, since 2009) scale and thus report Absolute Salinity in g/kg. However, given that global datasets such as GLODAP and WOA report salinity on the PSS-78 scale, calculated from conductivity, temperature and pressure, clarification in the text that values are dimensionless and follow PSS-78 are sufficient.

We have corrected the terminology to avoid using "PSU" when referring to salinity values. A note has been added beneath Table 1 clarifying that all salinity values follow the Practical Salinity Scale (PSS-78) and are unitless.

**Detailed comments:**

**Line 13, 14:** The study would benefit from more cautious wording, as it presents no direct evidence for changes in the carbon storage capacity.

We thank the reviewer for this suggestion. We have revised the abstract to adopt more cautious wording and to avoid implying direct evidence for changes in deep-ocean carbon storage. Specifically, the discussion of carbon storage has been reframed as being consistent with existing hypotheses and prior studies, rather than demonstrated directly by our data. The abstract now emphasises that our results support a physical mechanism that may contribute to the MPT, while acknowledging that confirmation awaits independent evidence from ice cores (e.g., Beyond EPICA–Oldest Ice).

The abstract now states that enhanced abyssal stratification is "consistent with previous hypotheses and data supporting increased glacial carbon storage," rather than asserting a direct causal relationship.

**Line 23:** One of the main conclusions of the study concerns a salinity change across the entire deep Pacific after 900 ka based on two Sites. However, the text states that $\delta^{18}O_{sw}$ only reflects variations in global ice volume and **local** hydrographic effects. This phrasing is somewhat ambiguous and should be clarified.

We have revised the text to more explicitly distinguish between the global and local components of $\delta^{18}O_{seawater}$.

The sentence now reads:
The interpretation of benthic $\delta^{18}O$ records is complicated, however, because the signal is dependent on both temperature and the oxygen isotopic composition of seawater ($\delta^{18}O_{seawater}$) which varies with global ice volume as well as regional hydrographic effects (Waelbroeck et al., 2002), such as salinity changes at sites of deep-water formation associated with changes in ocean circulation.

**Line 32:** This statement is too vague. Since this appears to be one of the main motivations of the present study, the divergent interpretations should be clearly introduced in the Introduction.

We have revised the Introduction to explicitly outline the divergent interpretations of Quaternary temperature and ice-volume evolution that motivate this study.

**Line 40-43:** Please specify what specific concerns are being referred to here and clarify whether "previous findings" refers to Yu & Broecker (2010) or Sosdian & Rosenthal (2009).

We have revised the text to identify the concerns raised by Yu and Broecker (2010) regarding the Mg/Ca-derived temperature record at Site 607 (Sosdian & Rosenthal, 2009), thereby clarifying the reference to "previous findings." Additional discussion of the carbonate ion effect remains in the Supplementary Information.

**Table 1:** Please use a consistent coordinate format. The authors should also cite the sources or datasets from which present-day temperature and salinity data were obtained.

All site coordinates have now been converted to decimal degrees, following the standard convention used in oceanographic databases.

We have also cited the sources and datasets from which present-day temperature and salinity values were obtained in Table 1.

**Fig. 1:** Please indicate in panel (a) where the transects shown in panels (b) and (c) were taken. Given potential differences between the eastern and western Atlantic, this spatial context is important. It would also be useful to include a single transect (covering the Pacific, Southern Ocean, and Atlantic) directly in the main manuscript rather than in the Supplement, as temperature gradients between these basins are a key aspect of the study.

We will revise Figure 1(a) to explicitly indicate the transect route shown in panel (b). The meridional profile shown in panel (c) was constructed as an idealised Pacific representation based on a very limited number of available deep ocean $\delta^{18}O_{seawater}$ estimates and does not correspond to a uniquely defined transect. We will clarify this explicitly in the figure caption and text, or to avoid potential confusion, we will either (i) revise panel (c) or (ii) remove this panel from the main figure and provide an indication of the transect route for Pacific temperature and salinity instead (i.e., panel (b) in Fig's. S1 and S2).

In response to the Reviewer's second point, we will move our meridional temperature profiles into the main manuscript to better illustrate large-scale temperature gradients that are central to our interpretation.

**Fig. 3:** Please use a consistent capitalisation for "Marine Isotope Stage" throughout the manuscript (either always "Marine isotope stage" as in line 81 or "Marine Isotope Stage," as in line 103). In addition, clarify the basis of the $2\sigma$ error bars mentioned in lines 82-83; are these derived from replicate measurements or represent analytical uncertainty or both?

We have ensured consistent capitalisation of "Marine Isotope Stage" throughout the manuscript. In addition, the figures have been revised to incorporate calibration uncertainties; accordingly, the

1σ and 2σ error bars now represent the combined calibration and analytical uncertainties and are not derived from replicate measurements.

**Line 89 and 92:** The phrase "and have similar seawater physical properties" is somewhat vague. Please specify which properties are referred to (e.g., temperature, salinity, density) or include a reference to Table 1 for clarification.

For clarification, we have included a reference to Table 1.

**Line 107-109:** Do all studies use the same approach for their age model?

The records compiled in this study do not all use identical age-models; however, they follow consistent and well-established approaches. All records, with the exception of MD99-2334 and MD01-2444 for the last 50 kyrs, are based on alignment of benthic foraminiferal δ¹⁸O to the LR04 reference stack or closely related reference stacks. MD99-2334 and MD01-2444 were independently tuned to ice-core chronologies. We now clarify the age-model approaches used for each record in the Methods section and note that, for the interval considered here, these chronologies are mutually consistent within age uncertainties. Accordingly, no additional retuning was applied.

**Suggested paragraphs:**
**2.1 Site details and chronology**
All previously published records used to compile the ocean stacks are presented on their original published age models. Most are based on alignment of benthic foraminiferal δ¹⁸O to LR04, including Deep Sea Drilling Project (DSDP) Site 607 combined with Chain 82-24-23PC 23PC (Ford et al., 2016; Sosdian and Rosenthal, 2009), and Ocean Drilling Program (ODP) Site 1123 (Elderfield et al., 2012). ODP Site 1208 (Ford and Raymo, 2020) was aligned to the Prob-stack using the HMM-Stack MATLAB code (Lin et al., 2014; Butcher et al., 2017; Ahn et al., 2017) which is effectively identical in structure to LR04 over the interval considered here. The ODP Site 1094 age model (Hasenfratz et al., 2019) was derived by graphical alignment of benthic δ¹⁸O to LR04, with the original chronology of Jaccard et al. (2013) retained where benthic foraminifera were sparse (e.g. MIS 4),

The Iberian Margin piston cores MD99-2334 and MD01-2444 were independently aligned to Greenland ice-core records through synchronisation of Dansgaard–Oeschger events recorded in planktonic δ¹⁸O (Shackleton et al., 2000; Skinner et al., 2003; Skinner and Elderfield, 2007). Due to the close agreement in both structure and absolute benthic δ¹⁸O values with Site U1385 over the interval of overlap, no additional adjustments to these published age models were applied.

[Figure]

Figure SXX: Benthic foraminiferal δ¹⁸O records from Iberian Margin piston cores MD99-2334 (orange line and circles)) and MD01-2444 (blue line and circles) shown on their original published age models, together with the benthic δ¹⁸O record from Site U1385 (black line with dark grey diamonds) for comparison. The records show close agreement in both structure and absolute values over the interval of overlap, including the timing of major glacial–interglacial and millennial-scale features. This close correspondence indicates that the published age models for the Iberian Margin piston cores are consistent with the Site U1385 chronology and do not require further adjustment.

**Line 122-123**: To my knowledge the MD01-2444 record spans ~50–35 ka, which does not include the LGM.

We have revised the text to clearly differentiate between the early (50–35 ka; MD01-2444) and late (35–10 ka; MD99-2334, including the LGM) parts of the last glacial period.

**Line 123-124:** Please Specify whether this offset is based on overlapping intervals between the cores or on modern temperature differences (which are not listed in Table 1).

Following Skinner et al. (2007), we applied a constant temperature offset of +0.6 °C to Mg/Ca-derived temperatures from the deeper Site MD99-2334 to account for depth-related cooling and to render the record comparable to Site U1385, and consistent with the nearby piston core MD01–2444. This offset is based on modern deep-water temperature gradients along the Iberian Margin as confirmed in depth-transect observations, ((Hodell, 2014; Skinner et al., 2003, 2007; Skinner and Elderfield, 2007); Table 1) rather than from overlapping downcore intervals. Modern temperature values supporting this adjustment have now been added to Table 1, and the methodological basis for the offset has been clarified in the text. The offset was applied within PSU Solver after temperature estimation and prior to the reconstruction of δ¹⁸O$_{seawater}$.

**Line 127:** I cannot fully understand this sentence. If the text refers to a specific taxonomic level, please use the term "species" to avoid confusion.

The sentence now reads:

"Measurements of benthic foraminiferal $\delta^{18}O$ at Site U1385, paired (same sample) with *Uvigerina peregrina* Mg/Ca analyses to reconstruct $\delta^{18}O_{seawater}$, were predominantly obtained from the same specimens of *Uvigerina peregrina* where sufficient calcite was available prior to cleaning for Mg/Ca analysis (Barker et al., 2003) (n = 560 representing 64 % of Mg/Ca samples)."

**Line 128:** Please clarify how comparable the results are between the two genera when both were measured.

We now include the following paragraph:
"Comparison of Mg/Ca–derived deep-water temperatures from *Uvigerina peregrina* and *Globobulimina affinis* at Site U1385 generally agree in both mean values and variability (Fig. 2a), with corresponding $\delta^{18}O_{seawater}$ estimates exhibiting consistent structure (Fig. 2b). This supports the species-specific Mg/Ca–temperature calibrations applied in this study (Elderfield et al., 2006, 2010, 2012; Weldeab et al., 2016) and justifies combining the two datasets into composite records of deep-water temperature and $\delta^{18}O_{seawater}$. The individual *Uvigerina peregrina* and *Globobulimina affinis* time series were used as inputs for the PSU Solver calibration procedures described in Section 2.4 and the combined records form the basis of the deep-water temperature and $\delta^{18}O_{seawater}$ reconstructions (Fig. 3) analysed in Section 3."

**Line 143-144:** Please specify which error the authors are referring to. No corresponding uncertainty is shown in Fig. 2a, so it is unclear what uncertainty estimate is referred to or how it was derived.

In the submitted version, the uncertainty referred to in lines 143–144 corresponds to analytical and replicate variability in the Mg/Ca-derived temperatures; however, this uncertainty was not shown in Fig. 2a, which we acknowledge was unclear. In the revised manuscript, all records will be reprocessed consistently through PSU Solver and the stacking procedure, and uncertainties will instead represent the combined analytical and calibration uncertainty. These uncertainties will be explicitly shown in Fig. 2a and described in the Methods and figure caption.

**Line 146:** It seems likely that the intended method was inductively coupled plasma–optical emission spectrometry (ICP OES), rather than "spectrophotometry".

Changed to spectroscopy.

**Line 166:** I think the intention was to refer to the excluded samples when using the term "the remaining samples." However, lines 166–169 read as if this refers to the samples actually used in the dataset. Please replace "remaining" with "excluded" to avoid confusion.

We have revised the text on Line 168 to clarify that the samples referred to are those retained in the final Mg/Ca dataset, and that sample exclusion was based on the identification of anomalously high co-occurring Fe/Ca–Mg/Ca values rather than the application of a fixed Fe/Mg threshold. We also clarify that the Fe/Mg values cited (<0.03 mol mol$^{-1}$) represent typical values reported in previous studies rather than exclusion criteria. The revised text now

explicitly distinguishes between excluded and retained samples, resolving the ambiguity noted by the Reviewer.

**Line 168:** Please remove the "<" before 0.03 mol/mol$^{-1}$ or instead use "< 0.1 mol/mol$^{-1}$," as Barker et al. (2003) only rejected samples with Fe/Mg < 0.1 mol/mol$^{-1}$, whereas the 0.03 mol/mol$^{-1}$ value cited here represents the typical range of their samples rather than an exclusion criterion.

See response for Line 166 above.

**Line 171:** units are missing.

Units have now been included.

**Line 172-177:** The reasoning in this paragraph is difficult to follow. The use of "rather" and "more likely" is confusing. "Ferromanganese oxide overgrowths" and "authigenic Mn–Fe-oxide coatings" refer to the same type of contamination.

The paragraph has been revised for clarity, consistency, and quantification. We now use a single term, "authigenic Mn–Fe-oxide coatings," to describe the observed Mn–Fe overgrowths, avoiding confusion caused by multiple phrases. The revised text also clarifies the interpretation of elevated Mn/Ca values and their implications for Mg/Ca.

**Line 174:** What is meant by elevated? Please quantify.

The term "elevated" is now quantified by explicitly providing Mn/Ca ranges for both *U. peregrina* (0.01–0.28 mmol mol$^{-1}$) and *G. affinis* (0.04–1.88 mmol mol$^{-1}$), with anomalous specimens noted. Mn/Ca values exceeding ~0.1 mmol mol$^{-1}$ are discussed in the context of potential authigenic Mn–Fe–oxide coatings.

**Line 175:** Please rephrase "this contamination" as not all Mn in foraminiferal calcite necessarily indicate contamination. It could also reflect (partly) natural incorporation of Mn during biomineralisation.

We clarify that higher Mn/Ca values do not necessarily reflect contamination alone; species-specific Mn incorporation during biomineralisation may also contribute (van Dijk et al., 2025). The absence of a positive correlation between Mn/Ca and Mg/Ca further supports that Mg/Ca variability is not significantly influenced by Mn–Fe oxides.

**Line 191:** This phrasing is ambiguous. Do the authors mean a 0.32 mmol/mol correction for Mg/Mn$_{coating}$ within formula 1, or a constant correction for all foraminiferal Mg/Ca data?

We have revised the text to explicitly state that 0.32 mmol mol$^{-1}$ refers to the Mg/Mn$_{coating}$ term used in Equation 1, rather than a constant correction applied directly to Mg/Ca values. This clarification has been added to the manuscript.

Text now reads:

To maintain consistency, we applied an Mg/Mn$_{coating}$ correction following Equation 1, with the Mg/Mn$_{coating}$ term fixed at 0.32 mmol mol$^{-1}$, and applied uniformly to all Mg/Ca data (Hasenfratz et al., 2017; de Lange et al., 1992) (Fig. S5).

**Equation 2:** How do the core-top Mg/Ca-derived temperatures compare with modern and/or pre-industrial deep-water temperatures at the study site? Please also justify the use of the Elderfield et al. (2012) calibration (originally introduced in Elderfield et al., 2010). The Elderfield et al., 2010 calibrations lower end is ~1°C. Roberts et al. (2016) specifically adjusted the Elderfield calibration for temperatures below 0 °C.

We apply the linear Mg/Ca–temperature calibration of Elderfield et al. (2012) for *Uvigerina peregrina* because it is species-specific, widely used in deep-sea Mg/Ca studies, and internally consistent with the calibration applied to the majority of records included in our regional stacks. Modern hydrographic data indicate that deep-water temperatures at all Iberian Margin sites examined here are ~2–3 °C (Hodell et al., 2014; Skinner et al., 2003). Consistent with these measured values, the most recent (core-top) portion of the Mg/Ca-derived temperature records shown in Figs. 2 and 3 yields temperatures of ~2.5 °C. As these values lie well within the calibrated range of the Elderfield et al. (2010, 2012) relationship, its application does not require extrapolation to sub-zero temperatures (Roberts et al., 2016). We therefore retain the Elderfield et al. (2012) calibration and clarify this in the Methods section.

**Line 225:** Is this uncertainty based on replicate measurements of the same standard material, or of multiple aliquots of the same sample?

The reported uncertainty is based on a pooled replicate error calculated from replicate analyses across multiple samples, rather than from repeat measurements of a single standard or multiple aliquots of the same sample. In this study, the pooled uncertainty is derived from 47 sets of replicate measurements and therefore reflects analytical reproducibility across the dataset as a whole.

**Figure 4:** Is there a reason why some interglacials and glacials are missing their labels in panel (a)? For example, MIS 13, 14, 17, and 18 are not indicated. Please clarify or add the missing MIS numbers for consistency.

We will ensure all MIS labels are added to this figure in the revised manuscript.

**Line 283:** Please ensure consistent use of time units throughout the manuscript. Earlier, "Ma/ka" was used for specific ages or time periods in the past, whereas "Myr/kyr" was used for durations. Here, "1.5 Myrs" is inconsistent with that convention. The same issue appears in line 285. Please standardise the terminology.

In both cases, we have revised the terminology to 'over the past 1.5 Myr,' replacing the previous usage of 'Ma'.

**Line 284 - 285:** This statement is difficult to follow. Is the observed cooling occurring during interglacials, glacials, or consistently across both climate states? Are uncertainties associated with this trend quantified? How are these numbers calculated?

In the revised manuscript, we will explicitly state whether the inferred cooling occurs during glacial, interglacial, or both climate states, and we will quantify the associated uncertainties. Cooling magnitudes at Sites 607 and U1385 will be recalculated using the revised datasets and described explicitly in the text, including the method used to derive these values.

**Line 313-316:** The comparison is difficult to follow, as it mixes different quantities: interbasin glacial–interglacial temperature differences for 1.5–0.9 Ma with glacial-only temperatures before and after 0.9 Ma. This makes the intended conclusion unclear. Please clarify the rationale for comparing these metrics and include uncertainties.

In the revised manuscript, we will ensure that like quantities are compared consistently, so that the intended conclusion is explicit. Uncertainties associated with these comparisons will be quantified and reported based on the revised data analyses.

**Line 319-321:** If the authors already write "We interpret the greater increase …," this should strictly not appear in the Results section.

We will move this interpretative statement from the Results to the Discussion in the revised manuscript.

**Figure 9:** The purpose of the dashed lines in Figure 9 is unclear. In panel 9b, it appears that the upper line might represent the mean interglacial $\delta^{18}O_{sw}$ across all interglacials and the lower line the maximum values—but this is not explained. Were these lines calculated from the data, or were they arbitrarily chosen? Similarly, in panel 9c, it is not evident how the dashed line is intended to represent "constant deep-ocean temperatures (line 343)." It appears to be placed at ~0 °C rather than reflecting a calculated mean. If this line is meant to represent an average, please show the associated variability (e.g., 2 SD).

The dashed lines in Figure 9 will be clearly defined on the revised Figure 9.

**Line 351-353:** Please add numbers for the Temperature decrease

We have added values for the decreasing temperatures and added more detail to the text.

**Line 357-360:** There may also be an important interbasin difference between the western and eastern North Atlantic, which challenges the assumption that Iberian Margin records can be treated as representative of the entire North Atlantic basin (e.g., Chalk et al., 2019). This aspect should be discussed, especially in relation to the statement in lines **287–290** that the carbonate ion effect cannot fully explain the temperature differences between Sites 607 and U1385. If the carbonate ion effect is not the sole driver of interbasin- and species Mg/Ca temperature offsets, the authors should explicitly consider and discuss alternative explanations, including east–west basin-scale hydrographic differences, which are currently not addressed anywhere in the manuscript.

We agree that potential east–west differences within the North Atlantic warrant more explicit discussion and represent an important limitation of treating the Iberian Margin as representative of the entire basin. In the revised manuscript, we will expand the discussion to address evidence for hydrographic differences between the western and eastern North Atlantic (e.g. Chalk et al., 2019, Hines 2024) and their implications for interpreting Mg/Ca-derived temperatures.

Furthermore, we will clarify that while the carbonate ion effect may contribute to the temperature offset between Sites U1385 and 607, it is unlikely to be the sole driver. We will therefore explicitly consider alternative explanations, including basin-scale hydrographic structure and differences in water-mass provenance. This will allow us to more clearly articulate both the assumptions and limitations associated with extrapolating Iberian Margin records to the wider North Atlantic.

**Section 4.1:** I would appreciate it if this entire section were given more structure. It is difficult to follow the connections between the different hypotheses and literature references and to identify the main message of the section. Throughout the paragraph, several mechanisms are mentioned (e.g., standing volume effect, NADW cooling, SCW salinity increase, ice sheet/shelf melting, and a stratification inversion in the North Atlantic), but they are not sufficiently integrated into a coherent narrative or overarching context in my opinion. For example, it is not clear to me whether the authors are arguing that the North Atlantic was bathed in NCW, in SCW, or favouring some other scenario.

Section 4.1 will be reorganised to explicitly separate competing circulation scenarios, outline their expected signatures, and state which interpretation is best supported by our data. In the revised section, we will more explicitly integrate the proposed mechanisms (e.g., standing volume effect, NADW cooling, SCW salinity changes, ice sheet and ice-shelf meltwater fluxes, and North Atlantic stratification changes) within a coherent interpretative framework. We will also make explicit which circulation scenario—whether a greater influence of NCW or SCW—we consider most consistent with our results, and clearly state the assumptions and limitations associated with this interpretation.

**Line 400:** Please remove strong, as it is relative.

Removed

**Figure 11c:** Please ensure that your own data points are clearly visible. They are currently obscured by the Grimmer et al. (2025) dataset, which prevents a proper comparison between the two records.

We will ensure our data points are visible in the revised figure.

**Line 425:** The period mentioned (2.75–2.15 Ma, according to the time frame defined in line 38) is not covered by the dataset. Please rephrase.

We recognise that the phrasing in Line 425 may have caused confusion regarding the temporal coverage of the dataset. To avoid ambiguity, we will rephrase this sentence to clarify that the interval discussed spans the onset of the MPT, rather than a period preceding it.

**Line 427-429:** Please clarify which MDOT dataset you are referring to and indicate the corresponding figure.

Done

**Line 431:** See comment on line 400

Done

**Line 431-433**: I would strongly recommend being more cautious with the argument that the records from Sites 607 and U1313 are complicated by multi-species measurements, since the record presented in this study (U1385) is also a derived from three different species.

We have revised the text to avoid implying that multi-species measurements uniquely complicate the records from Sites 607 and U1313. While the Site U1385 record is derived from two benthic species rather than three, we have removed the phrase "…complicated by multi-species measurements…" to ensure a more balanced and consistent treatment of multi-species records across sites.

**Figure 12:** Please indicate uncertainties

Figure 12 will be revised following completion of the new calibration analyses and will show uncertainties in the final revised version. Please see the response to anonymous Reviewer 1 point 3 in which a provisional Figure 12 is provided.

**Line 453-456:** Are there any similar approaches applied to more recent, data-rich intervals? A comparison would help assess uncertainties.

Similar tandem Mg/Ca–benthic $\delta^{18}O$ approaches have indeed been applied to a more recent, data-rich interval over the late Pleistocene and Last Glacial Maximum (Martin et al., 2002). Comparison with this study may demonstrate that increased site density improves constraints on both temperature and $\delta^{18}O_{seawater}$ and facilitate a more robust assessment of regional versus global signals. We will now explicitly reference this example–and any others obtained through a deeper literature search–in the discussion to place our results in the context of better-constrained intervals and to clarify that the primary limitation of our study arises from sparse spatial coverage rather than from the tandem proxy approach itself.

**Conclusion**: The conclusion includes a substantial amount of important and relevant literature. I would appreciate it if these key studies were introduced in more detail in the Introduction, allowing the conclusion to focus more clearly on the study's own findings.

As addressed in our revision of the Introduction (see Major Comment 5), key studies will now be introduced in more detail earlier in the manuscript. This allows the Conclusion to focus more clearly on the primary results derived from our stacked temperature and $\delta^{18}O_{seawater}$ reconstructions.

**Line 465-467:** Most of the studies cited in this context examine glacial–interglacial changes—often limited to the last glacial cycle—and do **not** address variability across the MPT. It is therefore unclear how these references support the conclusions drawn here or how they relate to the long-term patterns discussed in the study.

In the revised manuscript, we will prioritise references that explicitly address long-term variability across the MPT when drawing conclusions. Studies focused on glacial–interglacial variability over shorter timescales will be more clearly contextualised, or removed where they do not directly support the long-term patterns discussed.

**References cited in this review that do not appear in the manuscript:**

An, Z., Zhou, W., Zhang, Z., Zhang, X., Liu, Z., Sun, Y., Clemens, S. C., Wu, L., Zhao, J., Shi, Z., Ma, X., Yan, H., Li, G., Cai, Y., Yu, J., Sun, Y., Li, S., Zhang, Y., Stepanek, C., Lohmann, G., Dong, G., Cheng, H., Liu, Y., Jin, Z., Li, T., Hao, Y., Lei, J., and Cai, W.: Mid-Pleistocene climate transition triggered by Antarctic Ice Sheet growth, Science, 385, 560–565, https://doi.org/10.1126/science.abn4861, 2024.

Butcher, S., King, T., and Zalewski, L., Apocrita—High Performance Computing Cluster for Queen Mary University of London: London, Queen Mary University of London Technical Report, 2 p., https://doi .org/10.5281/ze- nodo.438045, 2017.

Chalk. T. B., Foster, G. L., Wilson, P. A..: Dynamic storage of glacial $CO_2$ in the Atlantic Ocean revealed by boron $[CO_2^{3-}]$ and pH records, Earth Planet. Sc. Lett., 510, 1–11, https://doi.org/10.1016/j.epsl.2018.12.022, 2019.

Clark, P. U. and Pollard, D.: Origin of the middle Pleistocene transition by ice sheet erosion of regolith, Paleoceanography, 13, 1–9, https://doi.org/10.1029/97PA02660, 1998.

Elderfield, H., Yu, J., Anand, P., Kiefer, T., and Nyland, B.: Calibrations for benthic foraminiferal Mg/Ca paleothermometry and the carbonate ion hypothesis, Earth Planet. Sci. Lett., 250, 633–649, https://doi.org/10.1016/j.epsl.2006.07.041, 2006.

Lamy, F., Winckler, G., Arz, H. W., Farmer, J. R., Gottschalk, J., Lembke-Jene, L., Middleton, J. L., van der Does, M., Tiedemann, R., Alvarez Zarikian, C., Basak, C., Brombacher, A., Dumm, L., Esper, O. M., Herbert, L. C., Iwasaki, S., Kreps, G., Lawson, V. J., Lo, L., Malinverno, E., Martinez-Garcia, A., Michel, E., Moretti, S., Moy, C. M., Ravelo, A. C., Riesselman, C. R., Saavedra-Pellitero, M., Sadatzki, H., Seo, I., Singh, R. K., Smith, R. A., Souza, A. L., Stoner, J. S., Toyos, M., de Oliveira, I. M. V. P., Wan, S., Wu, S., and Zhao, X.: Five million years of Antarctic Circumpolar Current strength variability, Nature, 627, 789–796, https://doi.org/10.1038/s41586-024-07143-3, 2024.

Li, K., Tian, J., Zhao, N., Du, J., Liu, Z., Du, J., and Huang, E.: Stable Pacific deep circulation punctuated by episodic intensification during the Mid-Pleistocene Transition, Glob. Planet. Change, 257, https://doi.org/10.1016/j.gloplacha.2025.105229, 2026.

Lin, L., Khider, D., Lisiecki, L. E., and Lawrence,  and C. E.: Probabilistic sequence alignment of stratigraphic records, Paleoceanography, 976–989, https://doi.org/10.1002/2014PA002713.Received, 2014.

McCave, I. N., Carter, L., and Hall, I. R.: Glacial-interglacial changes in water mass structure and flow in the SW Pacific Ocean, Quat. Sci. Rev., 27, 1886–1908, https://doi.org/10.1016/j.quascirev.2008.07.010, 2008.

McKay, R., Naish, T., Powell, R., Barrett, P., Scherer, R., Talarico, F., Kyle, P., Monien, D., Kuhn, G., Jackolski, C., and Williams, T.: Pleistocene variability of Antarctic Ice Sheet extent in the Ross Embayment, Quat. Sci. Rev., 34, 93–112, https://doi.org/10.1016/j.quascirev.2011.12.012, 2012.

Roberts, J., Gottschalk, J., Skinner, L. C., Peck, V. L., Kender, S., Elderfield, H., Waelbroeck, C., Vázquez Riveiros, N., and Hodell, D. A.: Evolution of South Atlantic density and chemical stratification across the last deglaciation., Proc. Natl. Acad. Sci. U. S. A., 113, 514–519, https://doi.org/10.1073/pnas.1511252113, 2016.

Scherrenberg, M. D. W., Berends, C. J., and Wal, R. S. W. Van De: $CO_2$ and summer insolation as drivers for the Mid-Pleistocene Transition, Clim. Past, 21, 1061–1077, 2025.

Whitworth, T., Warren, B. A., Nowlin, W. D., Rutz, S. B., Pillsbury, R. D., and Moore, M. I.: On the deep western-boundary current in the Southwest Pacific Basin, Prog. Oceanogr., 43, 1–54, https://doi.org/10.1016/S0079-6611(99)00005-1, 1999.

Willeit, M., Ganopolski, A., Calov, R., and Brovkin, V.: Mid-Pleistocene transition in glacial cycles explained by declining CO2 and regolith removal, Sci. Adv., 5, 1–9, https://doi.org/10.1126/sciadv.aav7337, 2019.

Wirths, C., Hermant, A., Stepanek, C., Stocker, T. F., and Sutter, J. C. R.: Sequence of abrupt transitions in Antarctic drainage basins before and during the Mid-Pleistocene Transition, Nat. Commun. , 16, https://doi.org/10.1038/s41467-025-65375-x, 2025.

---

## Author Comment (AC3)

In a paper published yesterday in this journal

Larsson, V. and Jung, S.: Persistent contamination in benthic-foraminifera-based Mg∕Ca thermometry using standard cleaning methods, Clim. Past, 21, 1871–1894, https://doi.org/10.5194/cp-21-1871-2025, 2025.

Present a different calibration for Uvigerina peregrina than the one used here (their Fig. 12) and some contamination issues are discussed. Please add some thoughts, if and how their findings are relevant for your study here.

Recent work has highlighted the sensitivity of *U. peregrina* Mg/Ca to residual contamination under certain cleaning protocols (Larsson and Jung, 2025); however, the stringent data quality evaluation employed in this study (Section 2.4) and the absence of covariance between Mg/Ca and Fe/Ca or Mn/Ca in our dataset indicates that contamination does not significantly influence the Mg/Ca-derived temperatures presented here.

The alternative calibration for *U. peregrina* shown in Fig. 12 of Larsson and Jung (2025) is based on earlier work by Elderfield et al. (2006) which relied on a relatively small number of core-top samples spanning ~1.5–12 °C. In contrast, the calibration of Elderfield et al. (2010, 2012) used in this study incorporates independent constraints on glacial-interglacial temperature change across the LGM-Holocene, yielding a Mg/Ca–temperature sensitivity of $\sim 0.1 \pm 0.013$ mmol mol$^{-1}$ °C.

Related discussion of the calibration choice and its applicability at low temperatures is provided in our response to Reviewer 2 (Equation 2).
* * *
**References cited here that do not appear in the manuscript:**

Elderfield, H., Yu, J., Anand, P., Kiefer, T., and Nyland, B.: Calibrations for benthic foraminiferal Mg/Ca paleothermometry and the carbonate ion hypothesis, Earth Planet. Sci. Lett., 250, 633–649, https://doi.org/10.1016/j.epsl.2006.07.041, 2006.

Larsson, V. and Jung, S.: Persistent contamination in benthic-foraminifera-based Mg/Ca thermometry using standard cleaning methods, Clim. Past, 21, 1871–1894, https://doi.org/10.5194/cp-21-1871-2025, 2025.